# Optimized backoff scheme for prioritized data in wireless sensor networks: A class of service approach

Innocent Uzougbo Onwuegbuzie[1,2]*, Shukor Abd Razak[1], Ismail Fauzi Isnin[1], Tasneem S. J. Darwish[1], Arafat Al-dhaqm[1]

1 School of Computing, Faculty of Engineering, Universiti Teknologi Malaysia, Skudai, Johor, Malaysia,
2 Department of Computer Science, The Federal Polytechnic Ado-Ekiti, Ekiti State, Ado-Ekiti, Nigeria

* iuonwuegbuzie@graduate.utm.my

**Data Availability Statement:** All relevant data are within the manuscript and its Supporting Information files.

## Abstract

Data prioritization of heterogeneous data in wireless sensor networks gives meaning to mission-critical data that are time-sensitive as this may be a matter of life and death. However, the standard IEEE 802.15.4 does not consider the prioritization of data. Prioritization schemes proffered in the literature have not adequately addressed this issue as proposed schemes either uses a single or complex backoff algorithm to estimate backoff time-slots for prioritized data. Subsequently, the carrier sense multiple access with collision avoidance scheme exhibits an exponentially increasing range of backoff times. These approaches are not only inefficient but result in high latency and increased power consumption. In this article, the concept of class of service (CS) was adopted to prioritize heterogeneous data (real-time and non-real-time), resulting in an optimized prioritized backoff MAC scheme called Class of Service Traffic Priority-based Medium Access Control (CSTP-MAC). This scheme classifies data into high priority data (HPD) and low priority data (LPD) by computing backoff times with expressions peculiar to the data priority class. The improved scheme grants nodes the opportunity to access the shared medium in a timely and power-efficient manner. Benchmarked against contemporary schemes, CSTP-MAC attained a 99% packet delivery ratio with improved power saving capability, which translates to a longer operational lifetime.

## 1 Introduction

Distance and time are no longer barriers, as people and things now communicate with each other anywhere and anytime simply with the push of a button or gesture control. Living and non-living things tagged with small and sometime non-noticeable electronic devices called sensor nodes communicate wirelessly by relaying sensed data to either near or remote locations where the data or information is needed for further processing. The application of this new technology cuts across every aspect of life such as home, industrial, logistics, aviation, health, manufacturing, and military. This new paradigm is referred to as the Internet of Things (IoT) [1]. Wireless Sensor Networks (WSN) [2, 3] forms the building block of IoT. WSN is

**Funding:** This research received funding with details below: Funder: Universiti Teknologi Malaysia (UTM) Award/Grant No: Q. J130000.2513.20H60 Awardee/Recipient: Shukor Abd Razak Funders URL: https://www.utm.my The funder had no role in study design, data collection and analysis, decision to publish, or preparation of the manuscript.

**Competing interests:** The authors have declared that no competing interests exist.

made up of tiny resource-constrained sensor devices called nodes or motes [4, 5]. Functioning cooperatively, these nodes are deployed in target locations to sense changes in the physical parameters of the target environment. Nodes in WSN are managed and controlled by a super-node mostly called the sink [6, 7], which in turn relays all gathered data to locations that may be within or completely outside the perimeter of the sensor network. The protocol that controls and regulates the functioning of WSN is the IEEE 802.15.4 standard [8, 9]. To attain a prolonged operational lifetime, the constrained power of sensor nodes must be carefully utilized as they are mostly autonomously powered by a battery.

While WSN is typically a data sensor, aggregator, and transportation network system, not all data traversing the network have the same class of service traffic priority [10, 11]. However, the standard IEEE 802.15.4 protocol which manages the operations of nodes is centered on an energy-efficient operation with no attention paid to data prioritization [9, 12], as data streaming through the sensor network are considered to be homogenous. This is mostly not the case in real life as most WSN deployments generate heterogeneous data of varying priorities. This inadequacy raises the question, how are data that needs urgent attention, such as time-sensitive mission-critical data, handled? What mechanism is in place to distinguish and identify various priorities of data streaming through the sensor network and how are these data processed to attain energy-efficient operation? Some deployment of WSN is intended to handle real-time, mission-critical data which may be time-sensitive, while others manage non-real-time data. In like manner, other deployment handles heterogeneous infrastructure, which generates both real-time and non-real-time data. The mechanism of how the IEEE 802.15.4 standard engaged in data propagation is discussed in the following section.

## 1.1. Carrier sense multiple access with collision avoidance (CSMA/CA) and superframe structure

With the Industrial, Scientific, and Medical (ISM) bands, IEEE 802.15.4 operates at the 2.4GHz frequency with a data rate of 250kbps [4]. Carrier Sense Multiple Access with Collision Avoidance (CSMA/CA) mechanism is implemented in two ways; Slotted and Non-slotted CSMA/CA. Using best effort (collision avoidance), the slotted mechanism grants node unguaranteed shared access to the transmission medium using superframe (see Fig 1), which is generated by the Personal Area Network (PAN) Coordinator (sink).

The superframe structure is divided into Active Period and Inactive Period [13, 14]. The active period is further divided into 16 time-slots which are shared by Contention Access

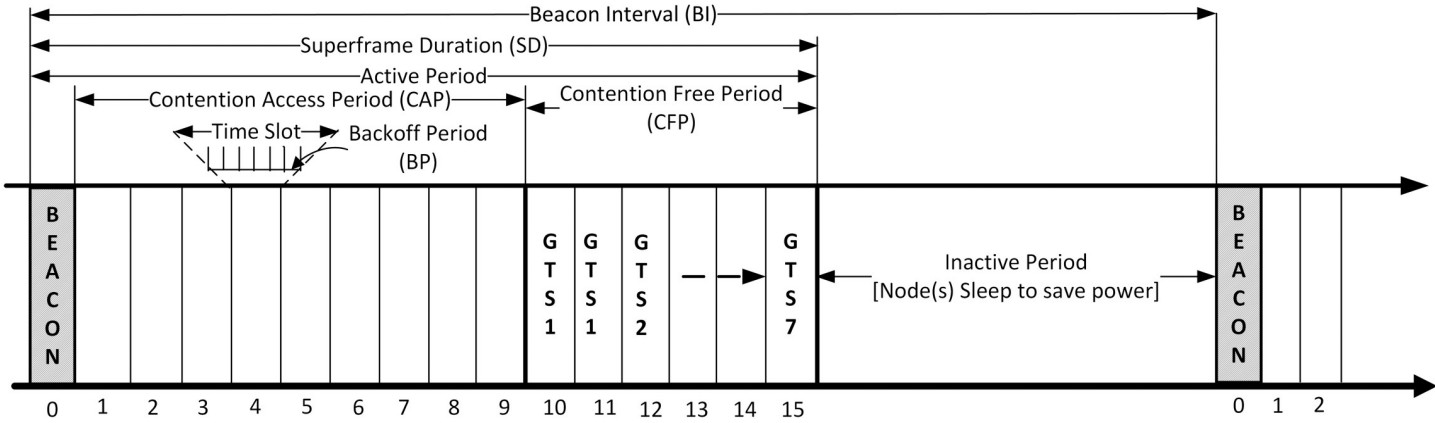

**Fig 1. IEEE 802.15.4 superframe structure.**

Period (CAP) and Contention Free Period (CFP) (the CFP may sometimes be optional), while the inactive period is used by nodes to conserve energy when nodes are not undergoing any active transactions [15, 16]. The 16 equally spaced time-slots run from zero to fifteen (0–15 time-slots), where zero (0) indicates the first time-slot [4]. The duration of a unit time-slot is equivalent to 0.96ms [17]. The first slot (slot 0) is used by the PAN Coordinator to broadcast the beacon frame [18], while the remaining 15 slots are used by the nodes to contend for channel access. The Beacon Interval (BI) indicates the interval between two successive beacon frames, while the length of the active period is indicated by the Superframe Duration (SD). BI and SD are obtained by two parameters; the Beacon Order (BO) and the Superframe Order (SO) (see Eqs 1 and 2), while the minimum superframe duration (*aBaseSuperframeDuration*) is represented by Eq 3. *aBaseSlotDuration* and *aNumSuperframeSlots* are constants provided by the IEEE 802.15.4 standard. *aBaseSuperframeDuration* is equal to the number of time-slots/ duration or symbols forming a Superframe when SO = 0 (see Eq 3). The duration is given as 960 Symbols [where 1 Symbol = 4 Bits = 16 microseconds (μs)], which corresponds to 15.36ms [19].

The CFP consists of Guaranteed Time Slots (GTS). These slots are not contended, here, nodes are given dedicated time-slots to carry out transactions (transmission and reception). The number of GTS is decided by the PAN Coordinator (does not exceed 7 slots) [15]. Requests for GTS is placed by any node that wants to execute a dedicated transaction without channel contention. Contrary to fixed time-slots awarded to nodes in CAP, in CFP, one GTS may take more than one time-slot (one GTS may be up to 2, 3, or even 4 time-slots as the case may be).

$$BI = aBaseSuperframeDuration \times 2^{BO} \tag{1}$$

$$SD = aBaseSuperframeDuration \times 2^{SO} \tag{2}$$

where; $0 \leq BO \leq 14$ and $0 \leq SO \leq BO \leq 14$

$$aSuperframeDuration = aBaseSlotDuration \times aNumSuperframeSlots \tag{3}$$

Slotted CSMA/CA uses the Binary Exponential Backoff (BEB) algorithm to compute a common range of backoff duration, which is the waiting time a node exercises before attempting to access the shared medium of transmission to avoid data collision. The problem here is that irrespective of the priorities of data, nodes randomly select the backoff period (BP) from a common pool of backoff range. However, the success of the transmission is not guaranteed as this sometimes leads to a collision, and the transmission is repeated. This repetitive process gradually and untimely degrades the node's scarce power leading to a shortened operational lifetime.

In the literature, some existing data prioritization schemes have attempted to proffer solution to this issue [9, 14, 20–22]; however, higher and lower priority data uses the same backoff computation algorithm or expression to generate the range of backoff periods (BPs) for repeated transmission attempt by the nodes, which causes the nodes to select the same BP for every failed transmission attempt, leading to unnecessary power consumption. In this article, we proposed the *Class of Service Traffic Priority MAC (CSTP-MAC)* scheme. In this scheme, nodes carrying varying priorities of data are classified according to the importance of their data. Instead of using a common backoff algorithm to compute BP, a set of unique backoff expressions peculiar to the data priorities are used to compute the range of BPs. By this, nodes are constantly provided with a non-conflicting range of BPs to choose from regarding their priorities, thereby granting them uncontested access to the shared channel of transmission.

The organization of this article is as follows; motivation and related work are discussed in Section 2, and Section 3 presents the methodological approach and design of the CSTP-MAC scheme, followed by Section 4 which talks about performance evaluation. Section 5 presents results and discussion, and the article ends with conclusions, limitations, and future work in Section 6.

## 2 Motivation and related work

The IEEE 802.15.4 standard provides no platform to address data heterogeneity or data priority [9, 12]. One of the crucial operational requirements of WSN is to handle data of different heterogeneity like real-time and non-real-time data. More often than not, WSN generates multiple streams of data traffic, which converges at a single device (sink) with changing speed and capacity. For instance, in the application of Wireless Body Area Network (WBAN) in the field of medicine, where Biomedical Sensor are implanted in the body of a patient to monitor blood pressure, body temperature, heartbeat rate, body oxygen level amongst others, data generated by these various body parts are prioritized according to the patients' medical condition, where time-constrained data are reported in real-time and classified as HPD, while delay-tolerant data are classified LPD and reported as non-real-time data. This same prioritization approach applies to other WSN deployments such as home, agriculture, logistics, and industry where data prioritization plays a major role. For timely and efficient processing of data with varying priorities, it becomes imperative to provide a data prioritization scheme by improving upon the IEEE 802.15.4 data handing mechanism to deal with data of varying priorities [18, 23]. Without prioritization, traffic with low data priority such as non-real-time data may dominate and congest the shared transmission medium with high bandwidth utilization, thereby hindering traffic with high data priority such as real-time data that may be carrying time-constrained mission-critical information. However, the standard IEEE 802.15.4 MAC scheme was designed with a focus on improving energy consumption and efficiency [22] without consideration for data priority [12], subsequently, the literature has not been able to proffer lasting solutions to this issue. To achieve data prioritization, this article adopts the concept of Class of Service (CS). CS is a traffic management service differentiation technique used to handle heterogeneous data or traffic, it classifies or groups services such as traffic or data having similar characteristics or features into unique levels of service priority [10, 11]. The implementation of this technique is explained in a later section.

### 2.1 Related work

To conserve the energy of constrained devices, IEEE 802.15.4 standard adopts CSMA/CA to check the activity state of the propagation medium before engaging in possible data transactions [20, 24] while bearing in mind how expensive it is to waste energy due to packet collision leading to retransmission. Before engaging in data transactions, in the slotted mechanism, nodes randomly select BP from the range of available periods computed by the BEB algorithm, and the countdown time is initiated. At the expiration of the countdown time (backoff timeslot or waiting time), the node attempts to gain access to the shared medium for data transmission. All nodes requesting to engage in network activities must align with the unit BP back to back. The timing operation of an ongoing transaction must elapse before a next queued node begins its transaction. A unit BP is a constant given as *aUnitBackoffPeriod* = 20 Symbols equivalent to 0.32 millisecond (or 320μs), subsequently equivalent to 80 bits [9, 24]. This is the smallest possible time a node can wait before Clear Channel Assessment (CCA) [24, 25] is performed. BP holds a range of *Lower Limit* and an *Upper Limit* for which a node can randomly

choose from. The range of BP is computed by Eq 4;

$$BackoffPeriod(BP) = [0, 2^{BE} - 1] \times aUnitBackoffPeriod \tag{4}$$

In non-slotted CSMA/CA, the backoff duration of one node is completely independent of the other and does not use beacon nor superframe, but utilizes the BEB algorithm and *BP* computation mechanism to compute for BP (see Eq 4). The *Backoff Exponential (BE)*, *Contention Window (CW)*, and *Number of Backoffs (NB)* are very important parameters for CSMA/CA. BE is used to compute the upper limit of a BP which ranges between zero (0) and $2^{BE}-1$; that is, the lower limit is 0 and the upper limit is $2^{BE}-1$. After the computation, at random, a BP is selected between the lower and upper limits which correspond to the waiting time a node has to observe before executing CCA operation [26, 27]. According to the standard, the minimum and maximum value of BE ranges from *macMinBE* = 3 and *macMaxBE* = 5, respectively [16, 24]. CW represents the number of channel activity checks before a node can gain access to transmit data packet in the channel, which corresponds to the CCA. For slotted CSMA/CA, CW has a value of 2, which means that after the backoff/wait time is completed, the mechanism immediately runs CCA twice to ensure that the shared medium is free of any ongoing transaction. NB represents the number of repeated attempts by a node to gain access to the shared transmission medium in the event of an initially failed transmission by the standard, *macMaxCSMABackoffs* = 5 (that is NB ranges from 0 to 4, where the first trial value is set to zero (0) and the fifth trial value is set to four (4)). At every successful access to the shared medium/channel, the value of BE and NB are decremented by 1, while CW is set to zero (0). If, however, access to the shared channel of transmission is not granted, the values of BE and NB are incremented by 1, while CW defaults to 2 (that is CCA to be re-executed twice again).

Henna, S. et al. [20] proposed the Traffic Adaptive Priority-Based MAC (TAP-MAC). In this work, data classes were categorized into three; class 1 –emergency, class 2 –on-demand, and class 3 –normal. The highest priority of all three is class 1 data. To avoid the denial of channel access to class 3 traffic, the traditional Contention Access Period (CAP) was modified accordingly; CAP1 (for Class 1 and Class 2) while CAP2 (for Class 3). With varying node numbers and traffic load, the CAP1 and CAP2 adaptively adjust to grant the respective node access to the shared medium. However, the implementation of TAP-MAC hinders class 1 (emergency data) from having immediate access to the medium as lower traffic class (class 3) seem to have greater access to the shared medium. Even though results show improved latency, throughput, and energy consumption as compared to the traditional IEEE 802.15.4 MAC scheme, the inability for higher priority traffic class to gain better access to the shared medium is defeated and inappropriate.

eMC-MAC was proposed by Pandit, S. et al. in their work [21] for Wireless Body Area Networks (WBAN). Data packets are categorized into Urgent Packets (UP), Critical Packets (CP), Reliability Constrained Packets (RP), Delay Constrained Packets (DP), and Normal Packets (NP). The standard Superframe was modified accordingly to accommodate these classes of data. UP is granted the highest priority to access the GTS while CP and RP contend with one another for access. However, the practical application of pre-emption was not implemented as suggested by the proposed MAC as there is no proactive mechanism to stop or interrupt the transaction of lower priority data of DP and NP once they gain access to the shared medium, most especially in the presence of UP.

Priority-Based Load Adaptive MAC (PLA-MAC) [22] proposed by Anjum, I. et al. targeted at improving the performance of heterogeneous traffic using service differentiation QoS requirements for WBAN. The traditional superframe was modified accordingly to suit the following parameters; CW, CAP, CFP, Beacon, and Low Power Listening (LPL). The CFP section

of the superframe was modified to accommodate Emergency Transfer Slot (ETS) and Data Transfer Slot (DTS). While CAP is of free length, that of CFP adjust dynamically to meet requirements according to data priority. High priority data make use of ETS while low priority data make use of DTS. While high priority data (emergency data) undergoes transactions, low priority nodes save energy by entering sleep mode. However, the implementations do not provide a mechanism to interrupt the transaction of low priority data to give room for high priority data, a situation that causes delays, hence the purpose of data prioritization is defeated.

Rasheed, M.B. et al. proposed the implementation of PG-MAC [14]. Traffic classification is prioritized into Normal Data (ND), Emergency Data (ED), and Periodic Data (PD). ND and PD are transmitted during the CFP by assigned GTS. As with other schemes, the standard superframe was modified to accommodate the extended period with Emergency Data Transmission Time (EDTT) which handles data transmission. However, this implementation does not meet the expected need as all classes of node priority are made to choose from the same range of Backoff time-slot. This increases collision and results in data retransmission, which adversely affects the constrained power of the nodes. Subsequently, throughput is reduced and latency increased.

TCP-CSMA/CA was proposed by Farhan Masud et al. in their work [9]. Nodes/traffic are classified and prioritized in the order of data importance. Critical Traffic Class (CTC) is assigned the highest priority, followed by Reliability Traffic Class (RTC), Delay Traffic Class (DTC), and finally, Non-constraint Traffic Class (NTC), which is assigned the lowest priority traffic class. The standard BEB algorithm was modified to obtain a unique algorithm for backoff timeslots for each traffic class. Even though the results looked promising, the complexity of backoff slot computation posed high computational overhead on nodes, leading to increased energy consumption, reduced throughput, and increased latency.

As it appears, efforts made in the literature to proffer lasting solution to data prioritization issue of the IEEE 802.15.4 standard has yielded no lasting solution, which leaves an open research area as further discussed;

- current literature addressing MAC prioritization protocol is often not appropriate as they mostly assume the data generated by WSN are homogenous with ideal channel access. In contrast, this is not the case in real life.

- the usage of the same backoff computation methods exhibited in the literature creates unbalanced traffic, which tends to direct traffic to the node(s) with possibly high data generation. This causes rapid exhaustion of power of such node(s), leading to a truncated network and shortening of the entire network lifetime.

## 3 Class of Service Traffic Priority-based MAC–(CSTP-MAC)

This paper introduces the data prioritization scheme known as CSTP-MAC. The scheme improves upon the IEEE 802.15.4 standard by the implementation of data prioritization to cater for heterogeneous data. The scheme adopts the techniques of Class of Service (CS) [10, 11] for data class prioritization. The methodological approach to achieving data prioritization is illustrated in Fig 2 and this approach applies to both single-hop and multi-hop data propagation models as shown in Fig 3. Data packets are classified into two main classes: Real-Time (RT), with HPD, tagged with decimal value CS = 0 and Non-Real-Time (NRT) with LPD tagged with decimal value CS = 1 (see Table 1). The definition of real-time is subjective to the application and the end-user [28, 29]. Fig 3a and 3b depicts a simplified deployment of a single-hop and multi-hop WSN for real-time and non-real-time events [9, 16]. The single-hop propagation is implemented for WBAN, while the multi-hop propagation is implemented for

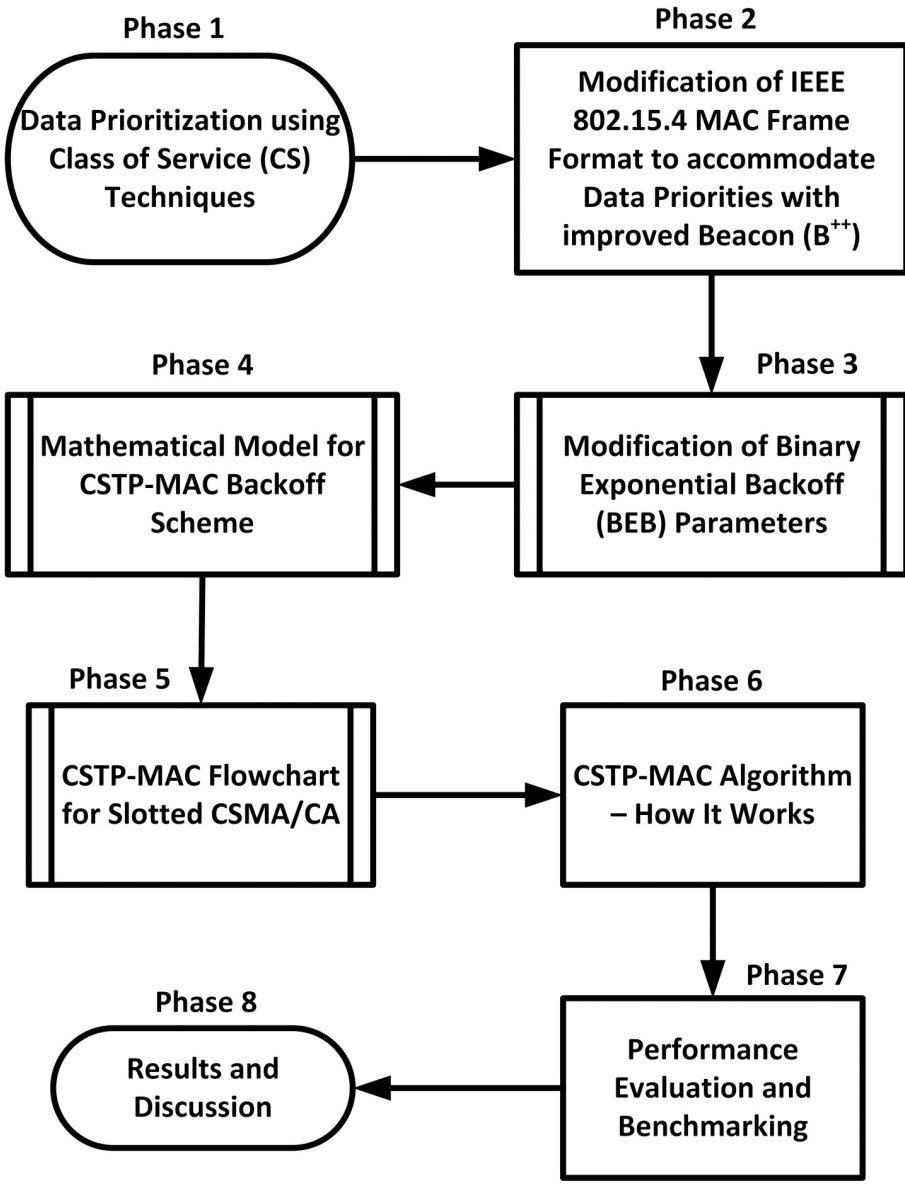

**Fig 2. CSTP-MAC methodological approach.**

other general applications like in home, industry, logistics, agriculture, and city, where data prioritization plays a deciding role. As shown in Fig 3a, for single-hop, at the discretion of the physician or medical expert, High Priority Nodes (HPN) are attached to patient body parts or organs where time-constrained HPD data are to be reported, while Low Priority Nodes (LPN) are attached to body parts or organs where delay-tolerant LPD data are to be reported. Subsequently, for Fig 3b with multi-hop propagation, nodes conveying similar priority of data are close to each other, where black coloured nodes A, B, C, D, E, F, G, and H are HPNs generating HPD, and green coloured nodes I, J, K, L, M, and N are LPNs generating LPD. Monitoring the target environment and implementing the CSTP-MAC scheme, heterogeneous stream of data generated by HPNs and LPNs are channelled to the Sink where further data processing is carried out.

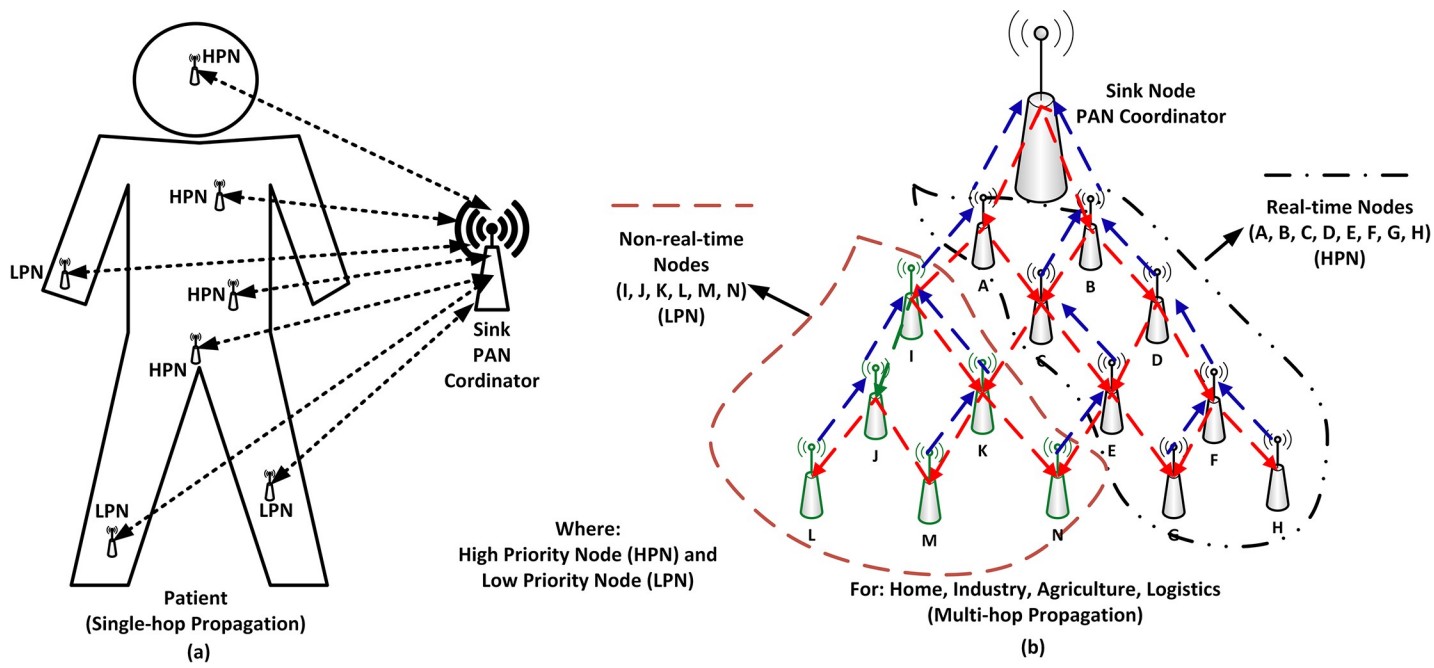

**Fig 3.** CSTP-MAC network topology (a) For the patient(s) in the medical field, and (b) For home, industry, agriculture, and logistics.

## 3.1 Modification of IEEE 802.15.4 MAC frame format–for CSTP-MAC data prioritization

The standard IEEE 802.15.4 MAC frame format was modified to accommodate the priority of heterogeneous data. Fig 4 shows a zoomed-in section of the *Frame Control* field, which contains the *Frame Type*, *Security Enabled*, *Frame Pending*, *ACK Request*, *PAN ID*, *Reserve*, *Destination Address Mode*, *Frame Version*, and *Source Address Mode* fields. With a focus on the *Frame Type* field, this field holds different types of frames generated by the IEEE 802.15.4 PAN Coordinator with their respective descriptions shown in Fig 4. *Beacon* Frame is in the memory address location of 000, *Data* Frame 001, *Acknowledgment (ACK)* Frame 010, *MAC Command* Frame 011 and finally, the *Reserve* field is between 100–111. To achieve data prioritization, the *Reserve* field under the *Frame Control* field was modified to accommodate the data prioritization implementation with the CS (shown with blue arrow), where data CS = 0 for HPD or CS = 1 for LPD, respectively. Subsequently, the MAC Control Part Sublayer (MCPS) data request primitive was also modified to capture data CS, where CS is indicated in blue underline as shown in Fig 5. These modifications change the standard *beacon* frame to what is referred to in this paper as *Beacon Plus Plus (B$^{++}$)* due to the inclusion of data priority class checks (see Fig 4).

### 3.1.1. Modification of binary exponential backoff parameters

The BEB algorithm is used in both slotted (beacon-enabled) and non-slotted (non-beacon-enabled) CSMA/CA to compute the range of backoff time-slots. This article discusses only the

**Table 1.  Traffic status and prioritization scheme.**

| Priority Status | Class of Service (CS) | Event Type |
|---|---|---|
| High Priority Data (HPD) | 0 | Real-time (RT) |
| Low Priority Data (LPD) | 1 | Non-real-time (NRT) |

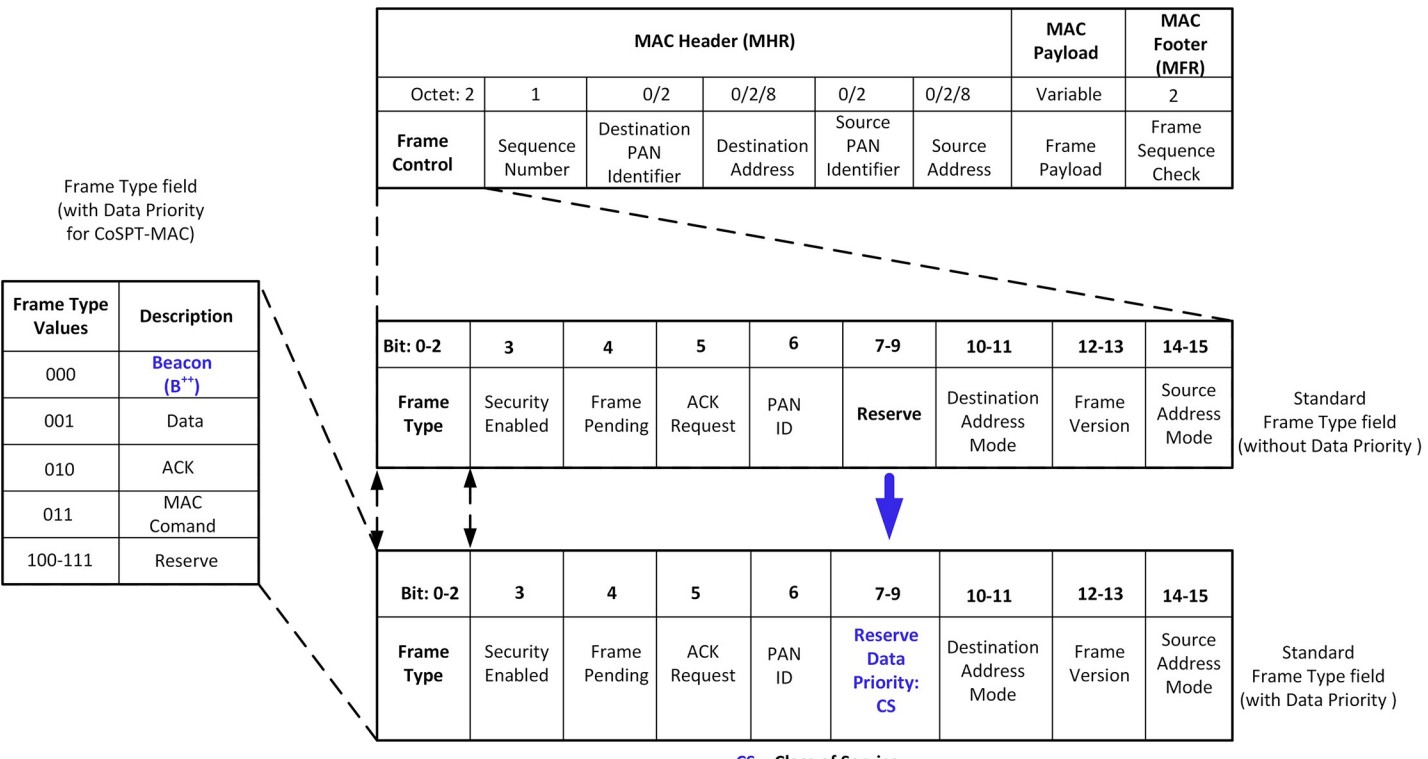

**Fig 4. Modified IEEE 802.15.4 MAC frame format (with data priority for CSTP-MAC).**

slotted beacon-enabled mode [9, 24]. In this mode, nodes randomly select BP or waiting time from the range of computed backoff time-slots using Eq 1, where $2^{BE}-1$ is known as the BEB algorithm. BE is bounded between $BE\{3,4,5\}$, where the minimum value of BE is 3 and its maximum value is 5. Computing possible backoff range with these limits $BE = 3,4$ *and* 5 will give a

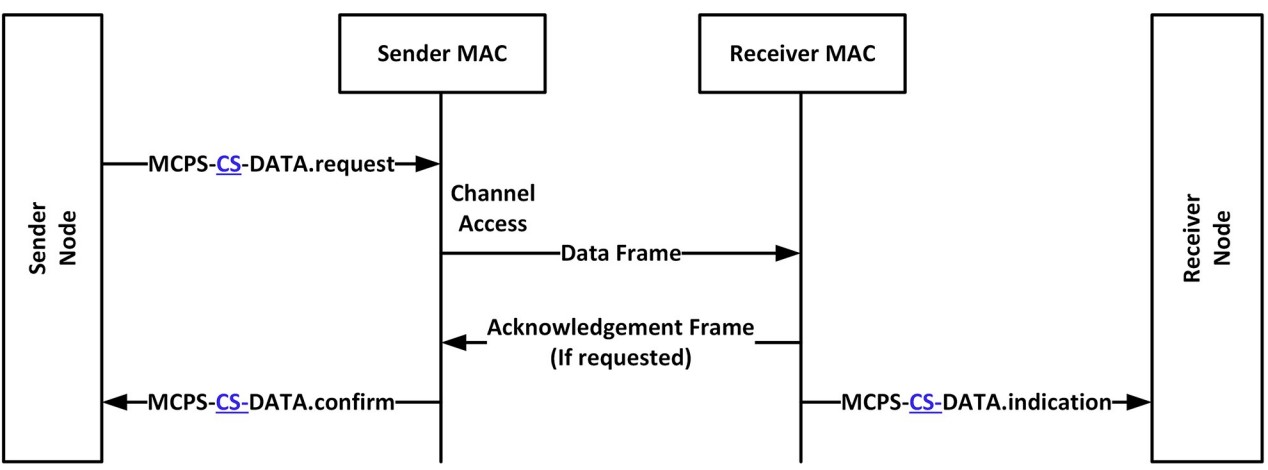

**CS** = Class of Service = Data Priority

**MCPS-CS-DATA.request(SrcAddrMode, SrcPANId, SrcAddr, DstAddrMode, DstPANId, DstAddr, msduLength, msduHandle, TxOptions, CS)**

**Fig 5. CSTP-MAC data service with data priority request.**

possible backoff range for $BE = 3;\{0,1,2.\ldots7\}$, $BE = 4;\{0,1,2.\ldots15\}$ and $BE = 5;\{0,1,2,.\ldots31\}$. Between either of these ranges, a node can randomly select BP. At the expiration of the waiting time, the node then attempts to engage in a network transaction. However, due to the rigidity of the backoff scheme, the above standard scheme failed to consider data of varying priorities as it uses a single algorithm (Eq 4) to compute BPs for all classes of data types (HPD and LPD as considered in this article). Additionally, this approach places an overhead on the network as the possible range of backoff time-slots grows exponentially as shown in Fig 6. This practice not only encourages network latency but leads to increased power consumption and shortened network lifetime. Mitigating against this shortcoming, the CSTP-MAC scheme uses a dedicated backoff computation algorithm peculiar to the data priority class to compute the range of possible backoff time-slots for HPD and LPD. To achieve this, the minimum and maximum values of BE are set to 1 and 5, respectively, as compared to 3 and 5 used by IEEE 802.14.5 standard (see Table 2). The CSTP-MAC backoff computation scheme is further discussed in sub-sections 3.3 and 3.4. The usage of these set parameters is shown in Table 3 of subsection 3.2

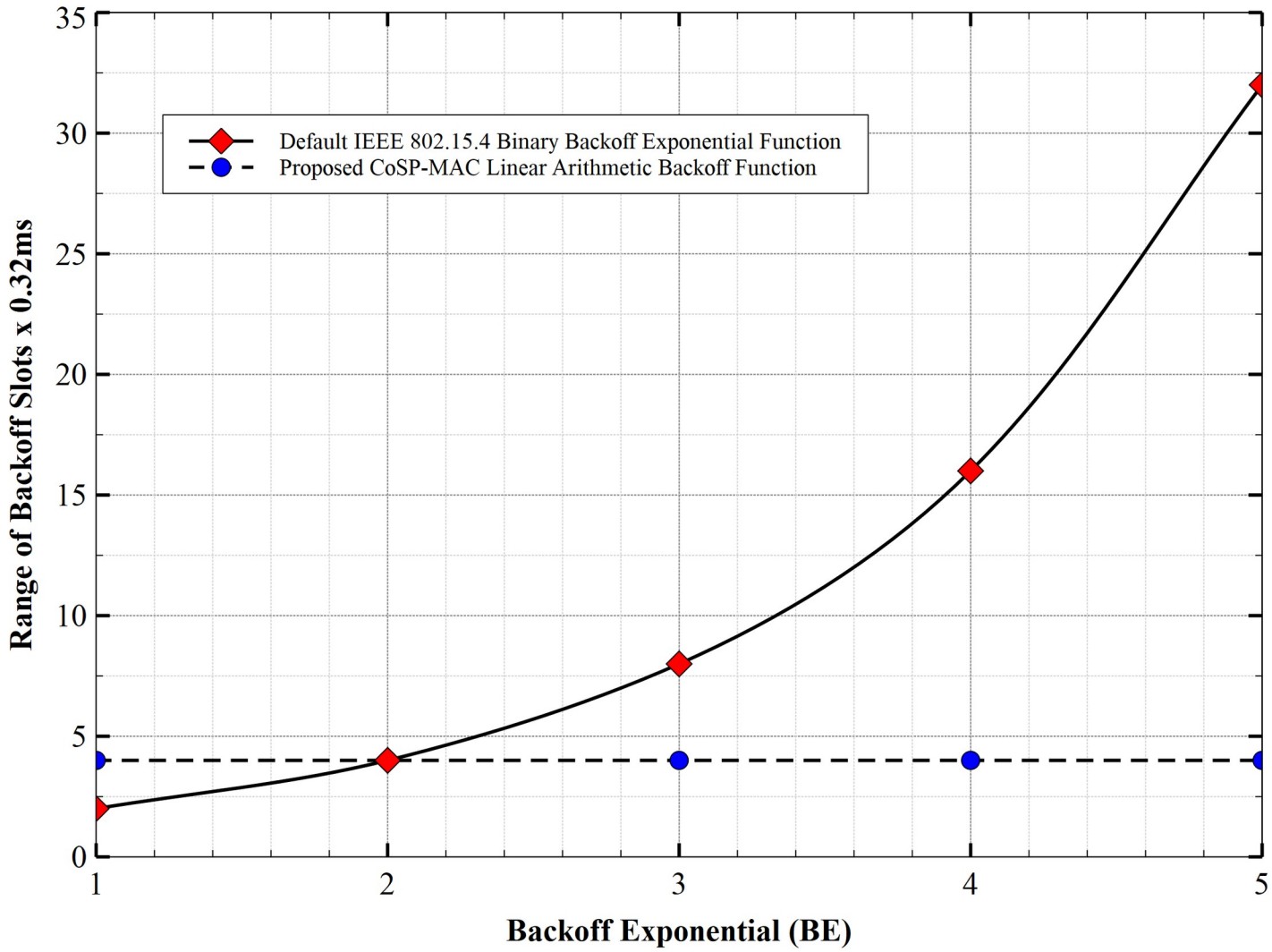

**Fig 6. CSTP-MAC algorithm backoff time-slots vs IEEE 802.15.4 binary backoff exponential algorithm backoff time-slots.**

**Table 2. Backoff Exponential (BE) parameters for BEB and CSTP-MAC for slotted CSMA/CA.**

| Binary Exponential Backoff (BEB) | | CSTP-MAC | |
|---|---|---|---|
| Parameters | Values | Parameters | Values |
| *macMinBE (BE)* | 1–3 (default 3) | *macMinBE (BE)* | 1 |
| *macMaxBE (BE)* | 1–5 (default 5) | *macMaxBE (BE)* | 5 |

## 3.2 The mathematical model for CSTP-MAC backoff scheme

In this article, we devised expressions to compute minimized backoff time-slots for different class of data in a relaxed and flexible manner. This is achieved by using uniquely different backoff time-slot estimators to compute the range of backoff time-slots for the lower to upper time limits for HPD and LPD respectively, without exceeding the maximum backoff time-slots obtained by the standard BP scheme, that is when *macMinBE* = 3 and *macMaxBE* = 5, which is 31 time-slot units. The backoff time-slots of CSTP-MAC is designed with the following conditions;

- maintain a steady arithmetic unit time growth with a fixed range of time-slots as compared to the exponentially increasing unit time-slots growth exhibited by the standard backoff range scheme

- define lower and upper backoff time limits; the difference between the upper and lower time limit is fixed at a maximum of 4 units.

- the decimal values of the data CS, that is CS equals 0 and 1 for HPD and LPD, respectively, are used as a multiplying factor to obtain distinct and unique expression to compute lower and upper time limits as shown in Eqs 5–9.

- the lower and upper backoff limit expressions are uniquely tied to a particular backup period computation (that is, either for 1st Backoff or 2nd Backoff or 3rd Backoff or 4th Backoff or 5th Backoff). In order words, a backup expression is not reusable to compute other backup rounds. This is contrary to the standard backoff computation practice, where the same back-off expression is used to compute all rounds of backoff periods.

**Table 3. CSTP-MAC algorithm mathematical models.**

| | | Lower Limits | Upper Limits |
|---|---|---|---|
| Eq 2 | | $CS2^{(BE+1)} +1$ | $2^{BE}+4CS+2$ |
| **First Backoff** | **CS = 0** | 1 | 4 |
| **BE = 1, NB = 0** | **CS = 1** | 5 | 8 |
| Eq 3 | | $(CS+2)2^{BE}-3$ | $2^{BE}+4CS+4$ |
| **Second Backoff** | **CS = 0** | 5 | 8 |
| **BE = 2, NB = 1** | **CS = 1** | 9 | 12 |
| Eq 4 | | $(CS+2)2^{BE}-4CS-7$ | $2^{BE}+4CS+4$ |
| **Third Backoff** | **CS = 0** | 9 | 12 |
| **BE = 3, NB = 2** | **CS = 1** | 13 | 16 |
| Eq 5 | | $2^{(BE-1)} + 4(CS+2) -3$ | $2^{BE} + 4CS$ |
| **Fourth Backoff** | **CS = 0** | 13 | 16 |
| **BE = 4, NB = 3** | **CS = 1** | 17 | 20 |
| Eq 6 | | $2^{(BE-1)} + 4CS+1$ | $2^{(BE-1)} + 4CS+4$ |
| **Fifth Backoff** | **CS = 0** | 17 | 20 |
| **BE = 5, NB = 4** | **CS = 1** | 21 | 24 |

- maximum backoff time should be less than 31 time-slot units

- the number of repeated backoff retries if a node fails to gain access to the shared transmission medium is 5 times, that is, the number of backoffs is set to $NB$:{0→4}; $macMaxCSMABackoffs$ = 4.

- minimum and maximum values of BE are set to $macMinBE$ = 1 and $macMaxBE$ = 5 respectively.

Satisfying the above conditions, the Eqs 5–9 are derived. The values or NB and BE are incremented by 1 until their maximum acceptable values are attained, which is 4 and 5, respectively, that is from the 1st to 5th backoff retries as shown in Eqs 5–9 (Table 4).

The computation of backoff time-slots with the above mathematical models to generate the units of upper and lower backoff time-slots limits of HPD and LPD is shown in Table 3. Using the mathematical models in subsection 3.2, it is shown that the maximum backoff time-slot for real-time events at CS = 0 is 20, which occurs at NB = 4 and BE = 5 (maximum allowable values), with lower limits ranging from 1 to 17 and upper limits ranging from 4 to 20 (that is from 1st backoff to the 5th BP attempts). Similarly, the maximum backoff time-slot for non-real-time events at CS = 1 is 24, which occurs at NB = 4 and BE = 5 (maximum allowable values), with lower limits ranging from 5 to 21 and upper limit ranging from 8 to 24 (that is from the 1st to the 5th BP attempts). Exceeding the NB and BE maximum allowable values triggers the report of failed transactions and the data/frame is dropped. With careful observation, it is shown that even for non-real-time events, the maximum backoff time-slot did not exceed 24 time-slot units compared to that computed by the BEB algorithm ($2^{BE}$−1) which reaches 31 (that is when $BE = macMaxBE$ = 5). Subsequently, with the CSTP-MAC model, the number of BP range for both RT and NRT events does not exceed 4 (that is between the higher and lower limits), which shows a linear arithmetic growth as compared to that obtained from the standard BEB algorithm, which grows exponentially with increasing BE values ($BE = macMinBE$ = 3and $BE = macMaxBE$ = 5) (see Fig 6). The implication of this is that the standard BEB introduces latency during the process of selecting BP from the exponentially increasing range of backoff slots, which is not an efficient mechanism for a resource-constrained device.

## 3.3 CSTP-MAC flowchart for slotted CSMA/CA

The operational dynamics of the CSTP-MAC scheme are presented in the flowchart diagram of Figs 7 and 8, where Fig 8 is a subpart of Fig 7 (presented in dashed rectangular lines) which represents the CSTP-MAC Backoff Period Computation Mechanism (CSTP-MAC–BPCM). CSTP-MAC computes unique backoff time-slots peculiar to the Class of Service of data traffic for $CS$ = 0;$HPD$ and $CS$ = 1;$LPD$, while the CSTP-MAC—BPCM mechanism computes distinct optimized and prioritized upper and lower limits of BP range for data priority to obtain a unique range of BPs. This process is iterated for BE {1, 2, 3, 4, 5} and NB {0, 1, 2, 3, 4}, equivalent to *1st Backoff*, *2nd Backoff*, *3rd Backoff*, *4th Backoff*, and *5th Backoff* stages for upper and

**Table 4. Eqs 5-9 shows the CSTP-MAC mathematical model for all five (5) backoffs.**

| No of Backoffs | Lower Limits | to | Higher Limits | |
|---|---|---|---|---|
| *1st Backoff* | $CS2^{BE+1}+1$ | | $2^{BE}+4CS+2$ | (5) |
| *2nd Backoff* | $2^{BE}(CS+2)−3$ | | $2^{BE}+4CS+4$ | (6) |
| *3rd Backoff* | $2^{BE}(CS+2)−4CS−7$ | | $2^{BE}+4CS+4$ | (7) |
| *4th Backoff* | $2^{(BE−1)}+4(CS+2)−3$ | | $2^{BE}+4CS$ | (8) |
| *5th Backoff* | $2^{(BE−1)}+4CS+1$ | | $2^{(BE−1)}+4CS+4$ | (9) |

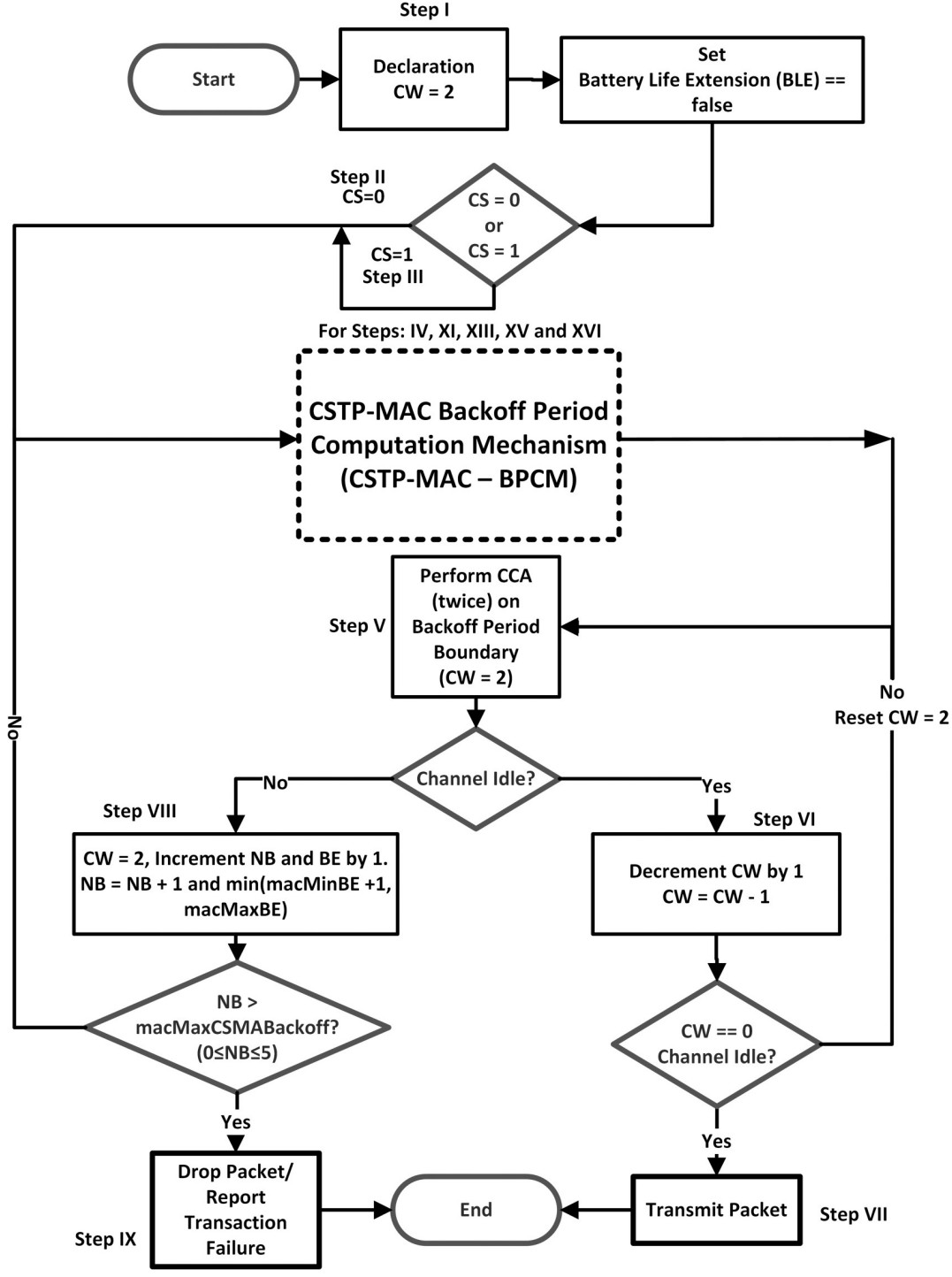

**Fig 7. CSTP-MAC flowchart for slotted CSMA/CA.**

lower limits of backoff time-slots. The scheme yields an arithmetically growing BPs that reduces the network latency and overall power consumption as compared to the standard IEEE 802.15.4 scheme that BPs grows exponentially. The resulting output of the CSTP-MAC scheme is shown in Table 3.

For Steps: IV, XI, XIII, XV and XVI

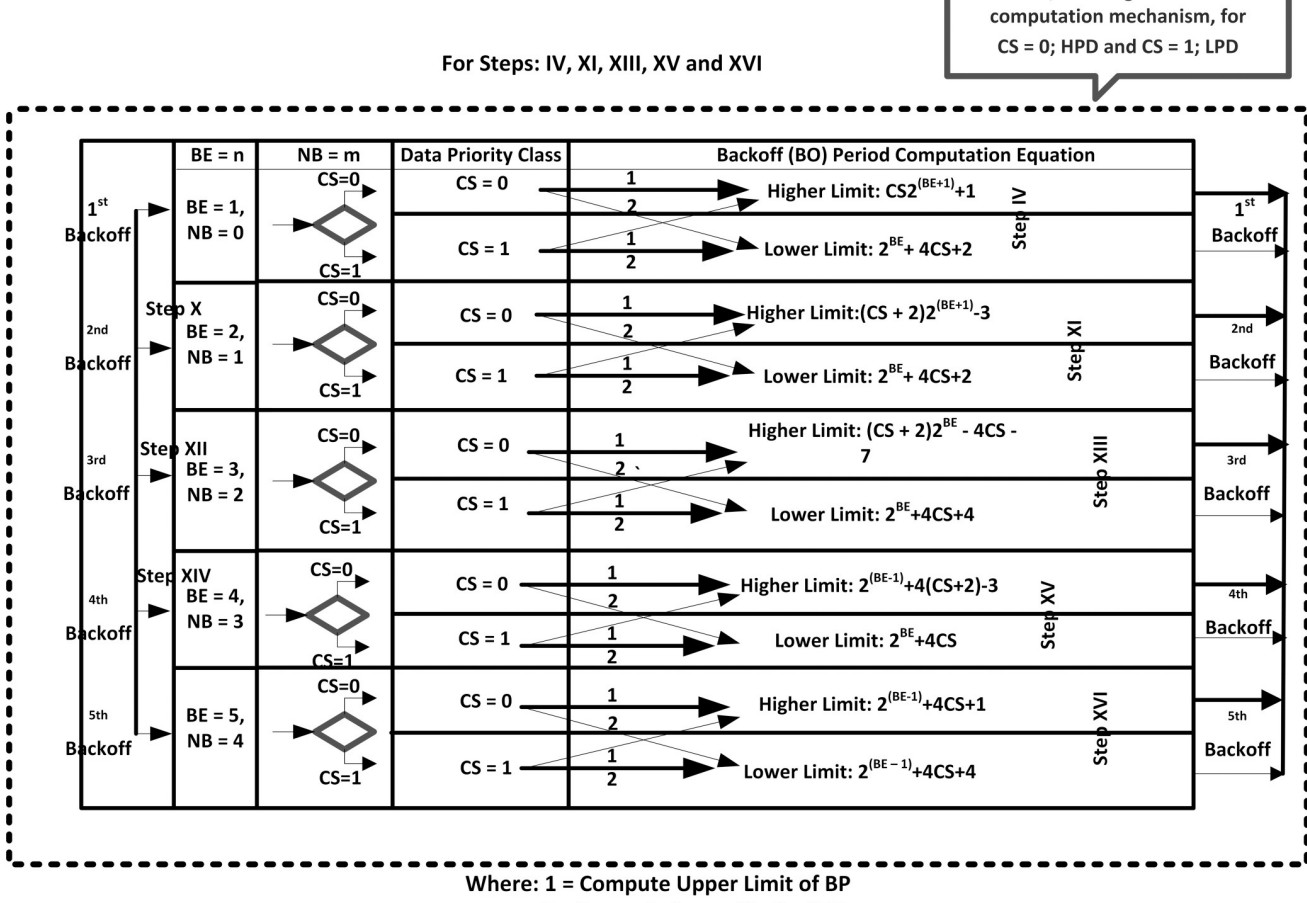

**Fig 8. CSTP-MAC backoff period computation mechanism (CSTP-MAC–BPCM).**

Fig 7 shows the flowchart diagram of how the CSTP-MAC scheme operates for a node intending to engage in a transaction. The operational workflow is explained in the steps below:

**Step I:** Declare $CW = 2$ and set Battery Life Extension (BLE) as false

**Step II and III:** Checks if $CS = 0$ or $CS = 1$ respectively and moves to **Step IV**

**Step IV:** Executes 1st Backoff attempt; set $BE = 1$ and $NB = 0$; if $CS = 0$, perform Higher Limit BP computation else perform Lower Limit BP computation and move to **Step V** up to **Step IX**; Iterate process for **Steps XI, XIII, XV,** and **XVI** for 2nd, 3rd, 4th, and 5th Backoff attempts respectively.

**Step V:** Perform CCA (twice) on the Backoff period boundary ($CW = 2$)

**Step VI:** Check Channel Idle? If Yes, decrement CW by 1 ($CW = CW−1$); check again if $CW == 0$, if Yes, move to **Step VII**, else increment $CW = 2$ and move to **Step V**

**Step VII:** Transmit Packet and end transaction

**Step VIII:** Check Channel Idle? If No, set $CW = 2$, increment NB and BE by 1 [$NB = NB+1$ and min($macMinBE+1$, $macMaxBE$)]. Check if $NB>macMaxCSMABackoff$? that is

($0 \leq N \leq 5$), if Yes, move to **Step IX**, else move to **Step IV** (perform next backoff attempts iteration)

**Step IX:** Drop packet and report failed transmission and end transaction

**3.3.1 CSTP-MAC network setup and initialization.** Network initialization and setup commences once nodes are powered on, and the sink sends the improved Beacon frame ($B^{++}$) across the network. As explained in subsection 3.1, $B^{++}$ contains the standard configuration, synchronization/management parameters, as well as the request for a node CS, which defines a node priority status. On receiving the $B^{++}$, nodes reply to the PAN Coordinator with their configuration settings (including CS priority status) and at the same time learns the configurations and priority status of neighbouring nodes. Using the mathematical model in subsection 3.2, following the data priority status, the range of backoff time-slots is computed for HPD and LPD.

## 3.4 CSTP-MAC algorithm–How it works

The CSTP-MAC Algorithm is presented in Algorithm 1. First, relevant/standard parameters are declared and initialized, the battery life extension (BLE) variable is set to false, CW = 2. For the start, the minimum value of $BE = macMinBE = 1$, and NB = 0. As explained in subsection 3.1, data class of service CS is identified through the improved MAC frame format to be either HPD or LPD, where CS is set to either 0 or 1. With NB = 0, the appropriate mathematical expression to compute the lower and upper limits of BP is selected and computed for the first backoff stage. After performing CCA twice, and the shared transmission medium is found not busy, the node transmits data, else, the attempt to gain access to the medium is repeated for the 2nd backoff until the 5th backoff stage (NB = 4), which is the maximum allowable attempt. With this improved channel access scheme of CSTP-MAC, each node can access the shared transmission medium with little or no contention, leading to reduced latency, high throughput, low power consumption with lowered radio duty cycle, high packet delivery ratio and prolonged network lifetime as demonstrated by the results shown in section 5.

```
Algorithm 1: The CSTP-MAC algorithm
Algorithm: CSTP-MAC: Class of Service Traffic Priority—MAC
 START
NOTATIONS:
High Priority Data HPD; Low Priority Data LPD.
 INPUT:
Backoff Exponent BE; Contention Window CW; Battery Life Extension BLE;
Clear Channel Assessment CCA; Number of Backoffs NB; Node Nd; Minimum
value of BE macMinBE; Maximum value of BE macMaxBE; Class of Service
CS; Maximum repeatable backoff macMaxCSMABackoffs
// Variables Declaration and Initialization
1. INIT BLE = false; CW = 2; NB = 0; macMinBE = 1: macMaxBE = 5;
macMaxCSMABackoffs
= 4 (i.e. 0≤NB≤4)
PROCESS
2. INIT CW ← 2, BLE ← false [step I]
3. //Compute and locate BP boundary (using the CSTP-MAC backoff
period computation algorithm)
4. IF Nd with CS = = 0 [step II]
//Execute First Backoff Stage for HPD
5. THEN INIT BE ← 1 and NB ← 0
6. GOTO [step IV]
7. ELSE
IF Nd with CS = = 1 then [step III]
```

```
//Execute First Backoff Stage for LPD
```
8. **INIT** CS ← 1
9. **GOTO** [step IV]
10. **ENDIF**
11. //Compute Lower Limit and Upper Limits with step 4, randomly choose backoff

between Lower and upper Limits and execute backoff duration

12. Delay by selecting random BP between $CS2^{(BE+1)}$ +1 (for CS = 0) to $2^{BE}$+ 4CS+2 (for CS = 1) **[step IV]**

13. Performs CCA with computed BP **[step V]**

14. **IF** Transmission medium = = idle

15. **THEN** INIT CW ← CW-1 **[step VI]**

16. **ELSE**

**IF** CW = = 0

17. **THEN** Transmit data **[step VII]**

18. **ELSE** Reset CW ← 2

19. **END IF** //Close inner IF

20. GOTO [step V]

21. **END IF** //Close outer IF

22. **ELSE**

**IF** (Transmission medium = = busy)

23. **INIT** CW ← 2, NB ← NB+1, BE ← min(macMinBE+1, macMaxBE) **[step VIII]**

24. **ENDIF**

25. **IF** NB > macMaxCSMABackoffs //i.e. 0≤NB≤5

26. **THEN** Nd drops packet, stop algorithm operation with message: Failed access

to transmission medium **[step IX]**

27. **ELSE**

**IF** BE = = 2 **[step X]**

28. //Compute Lower Limit and Upper Limits with step XI, randomly choose backoff

between lower and upper limits and execute delay

29. **THEN** INIT NB ← 1 and delay by selecting random BP between $(CS+2)2^{BE}$-3 (for CS = 0) to $2^{BE}$+ 4CS+4 (for CS = 1) **[step XI]**

30. GOTO [step V]

31. **ELSE**

**IF** BE = = 3 **[step XII]**

32. //Compute Lower Limit and Upper Limits with step 13, randomly choose backoff

between Lower and upper Limits and execute delay

33. **THEN** INIT NB ← 2 and delay by selecting random BP between $(CS + 2)2^{BE} - 4CS -7$ (for CS = 0) to $2^{BE}$+4CS+4 (for CS = 1) **[step XIII]**

34. **GOTO** [step V]

35. **ELSE**

**IF** BE = = 4 **[step XIV]**

36. //Compute Lower Limit and Upper Limits with step 15, randomly choose backoff

between Lower and upper Limits and execute delay

37. **THEN** INIT NB ← 3 and delay by selecting random BP between $2^{(BE-1)}$ +4(CS+2)-3 (for CS = 0) to $2^{BE}$+4CS (for CS = 1) **[step XV]**

38. GOTO [step V]

39. **ELSE**

40. //Compute Lower Limit and Upper Limits with step 16, randomly choose backoff

between Lower and upper Limits and execute delay

41. **THEN** INIT NB ← 4 and delay by selecting random BP between

```
   2^(BE-1) + 4CS+1 (for CS = 0) to 2^(BE- 1) +4CS+4 (for CS = 1) [step XVI]
42. GOTO [step V]
43. ENDIF //end inner if for varying values of BE [step XVII]
44. ENDIF //end outer if; checks if NB > macMaxCSMABackoffs [step
XVIII]
OUTPUT: Optimized prioritized data channel access to HPD and LPD to
shared
transmission medium with improved network performance and prolonged
network lifetime.
45. END
```

## 4 Performance evaluation

To validate the performance of the CSTP-MAC scheme, an extensive simulation was performed and performance comparison between HPD and LPD was carried out. Subsequently, the scheme was benchmarked against existing data priority-based MAC schemes; TCP-CSMA/CA [9], PLA-MAC [22], eMC-MAC [21], and PG-MAC [26]. The simulation is performed with Cooja [15, 30]; an event-driven WSN simulator/emulator bundled with the Contiki Operating System; a state-of-the-art operating system for IoT [25]. Incorporated within the Contiki OS is the Zolertia Z1 mote used as the nodes and run on the MSP430 microcontroller with CC2420 radio frequency (RF) transceiver [5, 31]. Contiki OS provides an efficient simulation environment for WSN with detailed statistical parameters. The Unit Disk Graph Medium (UDGM), with distance loss propagation model, was adopted with Constant Bit Rate (CBR) traffic model which is most suitable for resource-constrained devices generating data at a constant rate [20, 29] with maximum frame size set to 127 bytes, which include 25 bytes of MAC Header and 102 bytes of payload [9, 17] operating within the standard operational carrier frequency of 2.4 GHz and channel data rate of 250 kbps [4, 9] (see Table 5).

### 4.1 Simulation model and parameters

To ensure that results obtained are realistic and consistent, the simulation model and parameters are designed similarly to [9, 13, 32]. Star topology was adopted with one PAN

**Table 5. Simulation configuration and parameters.**

| Parameter | Value | Parameter | Value |
|---|---|---|---|
| Operating Carrier Frequency | 2.4 GHz | aMaxPHYPacketSize (MAC Frame Size) | 127 bytes |
| Channel Data Rate | 250 kbps | aBaseSlotDuration | 60 symbols |
| A Slot Duration | 15.36ms (960 Symbols) | aNumSuperframeSlots | 16 |
| Slot Mode | Beacon-Enabled | aMaxFrameRetries | 3 |
| CCA Duration | 8 symbols | Minimum BE (macMinBE) | 1 |
| Traffic Type | Constant Bit Rate (CBR) | Maximum BE (macMaxBE) | 5 |
| PAN Coordinator | 1 | Battery Life Extension (BLE) | False |
| Nodes | 30 and 14 | Simulation Time | 2000 s |
| Node Orientation | Random placement with no mobility | Operational Voltage | 3 Volts |
| Network Topology | Star | Current Consumption (OFF Mode) | <1μA |
| Tx/Rx Range | 50 x 50 m2 | Current Consumption (Power Down Mode) | 20μA |
| Battery Life Extension | False | Current Consumption (IDLE Mode) | 426μA |
| macMaxCSMABackoffs | 0–4 (5 retries) | Current Consumption (Rx Mode) | 18.8mA |
| aUnitBackoffPeriod | 20 symbols (0.32ms) | Current Consumption (Tx Mode @0dB) | 17.4mA |
| Propagation Model | Unit Disk Graph Medium (UDGM): Distance Loss | Transmission Support | Single Hop and Multi-hop |
| Node/Mote Type | Zolertia Z1 Mote with MSP430 Microcontroller Unit | RF Transceiver | CC2420 module |

Coordinator. For performance comparison between HPD and LPD, one hundred (100) nodes were used in a multi-hop propagation model, and for benchmarking with related schemes [9, 21, 22, 26], fourteen (14) nodes were used in a single-hop propagation model. Each simulation category was performed separately. Nodes are positioned randomly in a 100 by 100 square meter field for HPD and LPD comparison and 50 by 50 square meter field for benchmarking with related schemes. All nodes have a clear line of sight to the PAN coordinator with no hidden nodes. HPD and LPD performance comparison are comprised of HPNs and LPNs, oriented in such a way that nodes with similar priorities are closer to each other as depicted in Fig 3b for multi-hop propagation. Each simulation was performed for 2000 seconds. Readings were obtained with an increasing number of nodes starting with the first node until the maximum node count is reached for each simulation category. Each round of simulation was repeated five (5) times and the average reading is obtained. Table 5 shows the simulation configuration and parameters.

## 4.2 Performance metrics

The performance metrics used for performance comparison between HPD and LPD as well as benchmarking with priority-based MAC implemented schemes are discussed below:

• **Packet delivery ratio (PDR)**

PDR measures network reliability. It is the ratio of the total packets received at the destination node to total packets sent by the source node [9, 33]. This is expressed in Eq 10. A PDR of 100% is said to be a network of good performance. It is presented in percent (%);

$$PDR = \frac{Total\ packets\ received\ at\ destination\ node}{Total\ packets\ sent\ by\ source\ node} \times 100\% \qquad (10)$$

• **Power consumption**

This is the measure of total power consumed at the various operational states; transmit (Tx), receive (Rx), idle and sleep states [34, 35]. This is expressed in Eq 11. It is measured in milliwatt (mW) or watt (W).

$$Power_{Total} = Power_{Tx} + Power_{Rx} + Power_{mcu-idle/listen} + Power_{mcu-sleep} \qquad (11)$$

• **Throughput**

Throughput measures network reliability. It is the ratio of total packets received to how much time is spent to receive the packets. That is, it is the measure of how much data is transferred over a given time. High throughput is an indication of good network performance. This is expressed in Eq 12. It is measured in bytes per second or bits per second (bps) [16, 23, 36].

$$Throughput = \frac{Total\ packets\ received}{Time\ taken\ to\ receive\ packets} \qquad (12)$$

• **Latency**

Network latency is a measure of a network's responsiveness. It defines how much time it takes a packet to go from the source node to the destination node. The closer the value of latency is to zero, the better the network performance [27, 37]. This is expressed in Eq 13. It is measured in millisecond (ms) or second (s).

$$Latency = Time\ packet\ is\ received - Time\ packet\ is\ sent \qquad (13)$$

- **Radio duty cycle (RDC)**

RDC is an indication of how much time a node spends in its various operational states; transmit, receive, idle, and sleep. Mostly presented in percent, it is directly proportional to power or energy consumption; that is, high RDC indicates high power or energy-consuming operation hence reduced network lifetime, while low RDC indicates otherwise [2, 19, 38]. Duty Cycle is controlled by varying the values of SO and BO and is expressed by Eq 14.

$$DC = \frac{2^{SO}}{2^{BO}} \times 100 \tag{14}$$

## 5 Results and discussions

This section discusses the results obtained from performance comparison and benchmarking with related schemes with regards to performance metrics.

### 5.1 CSTP-MAC data priority classes performance comparison

To truly understand the impact of traffic prioritization with the CS approach, it becomes necessary to show in clear terms how this approach impacts on the network's performance concerning HPD–for real-time applications and LPD–for non-real-time applications.

**5.1.1. Packet delivery ratio (PDR)–HPD vs LPD.** PDR comparison of HPD and LPD is presented in Fig 9. The PDR of HPD and LPD remained consistent at 100% performance with

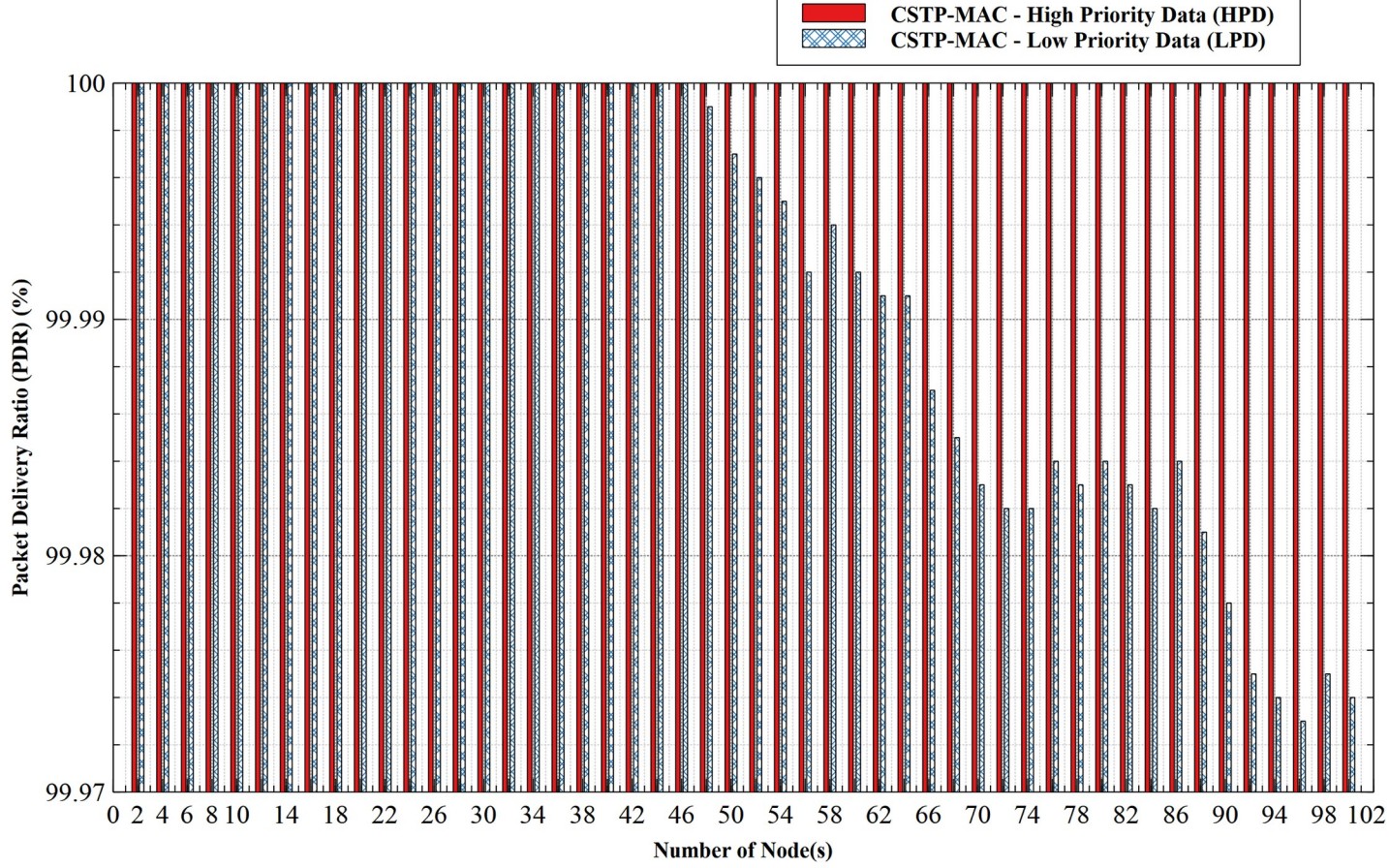

**Fig 9. Packet delivery ratio (PDR)–HPD vs LPD.**

an increasing number of nodes until 46 nodes. While that of HPD continued to remain constant at 100% till 100 nodes, that of LPD begins to drop at 48 nodes with the least performance of 99.973% at 96 nodes, which is still a good network performance indicator. The consistency of HPD PDR results from the fact that it is granted uncontested access to the shared transmission medium whenever high priority nodes have data to transact; subsequently, backoff time-slots are chosen from the lower limits of the range of available backoff slots for HDP compared to LPD whose lower limit backoff time-slot is much higher than that of HDP. This reduces or eliminates channel contentions between data of both CS, resulting in reduced latency and improved network PDR for HPD.

**5.1.2. Power consumption–HPD vs LPD.** Fig 10 shows the power consumption comparison between HPD and LPD under the same condition. The outright difference between data of both CS is revealed. HPD consumes much lesser power as compared to LPD. For the least power consumption, at 2 nodes, HPD consumes 0.592mW while LPD consumes 0.734mW. Growing consistently with an increasing number of nodes, the peak consumption for HPD and LPD reached 1.145mW and 0.747mW, respectively, at 100 nodes. Result also shows that the increase in HPD values is less aggressive as compared to LPD because data generated from high priority nodes are meant to choose their BPs from the lower limits of the available range of backoff time-slots. This greatly reduces the waiting time before accessing the channel, which

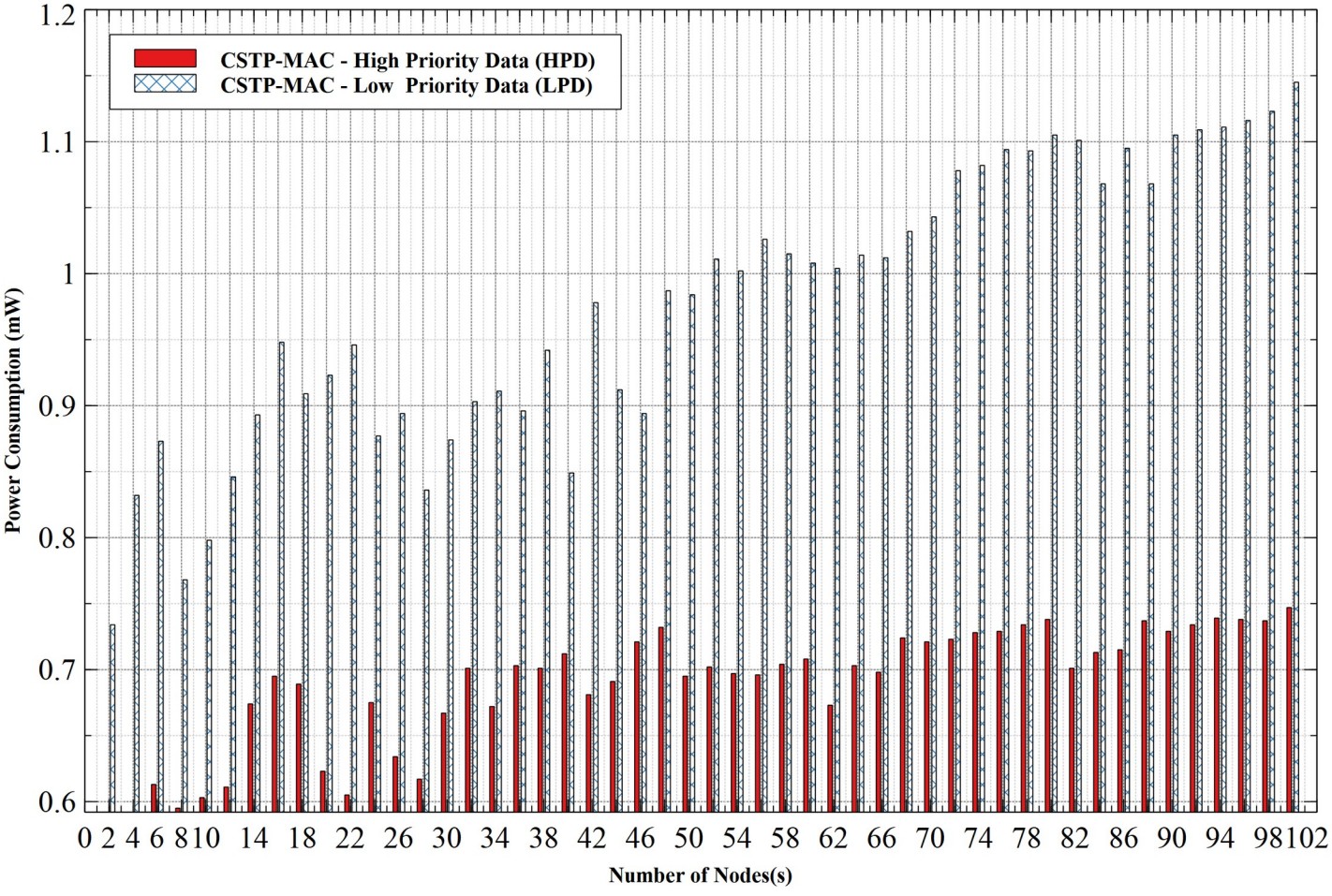

**Fig 10. Power consumption–HPD vs LPD.**

at this stage is without contention. On this ground, HPD effortlessly gains full access to the shared medium with the most reduced power utilization. The opposite of this is said of LPD.

**5.1.3. Throughput–HPD vs LPD.** Fig 11 exhibits the throughput comparison between HPD and LPD. The result and figure show that HPD consistently outperforms LPD with an increasing number of nodes. HPD attained maximum throughput of 35bps with a corresponding value of 32bps for LPD at 56 nodes. Due to a gradual increase in the collision domain with increasing nodes, resulting in the zig-zag patterned graph, the values or both data CS dropped gradually to values of 26bps and 21bps, respectively, at 100 nodes. The better performance exhibited by HPD results from its ability to gain uncontested access to the shared propagation medium with minimal backoff time-slots to compute its waiting time before accessing the shared medium. This action increases network performance with better throughput for HPD. In contrast, the lower backoff time-slots limit of LPD is considerably higher than that of HPD, resulting in longer waiting times before accessing the shared channel as compared to HDP. This encourages increased latency for LPD with reduced throughput measures.

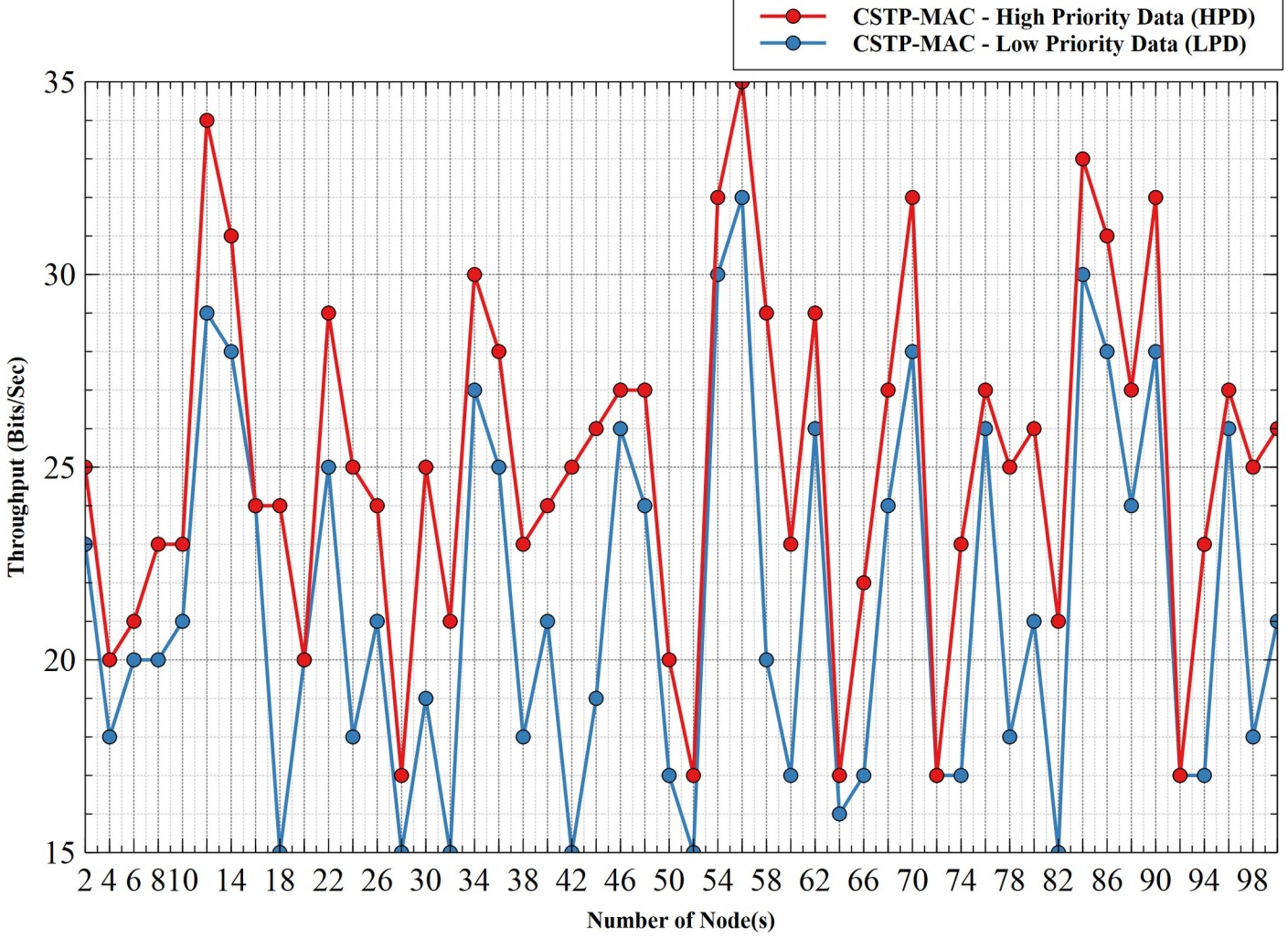

**Fig 11. Throughput–HPD vs LPD.**

**5.1.4. Latency–HPD vs LPD.** Fig 12 shows the latency comparison between HPD and LPD. Results show increasing latency values with an increasing number of nodes with HPD having better performance, with peak values of 66ms and 79ms for HPD and LPD, respectively, at 100 nodes. LPD attained higher latency compared to HPD, which is a result of an increase in the collision domain with an increasing number of nodes. Data generated by low priority nodes randomly select their backoff time-slots from the higher range of the available backoff slots to compute their waiting time. This accounts for increased waiting time and increased collision probability leading to high latency for LPD. The minimal value of backoff time-slots selected by HPD helps reduce waiting time with reduced network latency, leading to better network performance.

**5.1.5. Radio duty cycle (RDC)–HPD vs LPD.** Fig 13 shows the RDC comparison between HPD and LPD for listening and transmit operations. With minimum values of 0.64%—listen and 0.01%—transmit for HPD and 1.23%—listen and 0.04%—transmit for LPD, these values remain consistent across the increasing number of nodes until the 100 nodes. Figure distinctively shows how listening operation accounts for a higher duty cycle compared to transmit

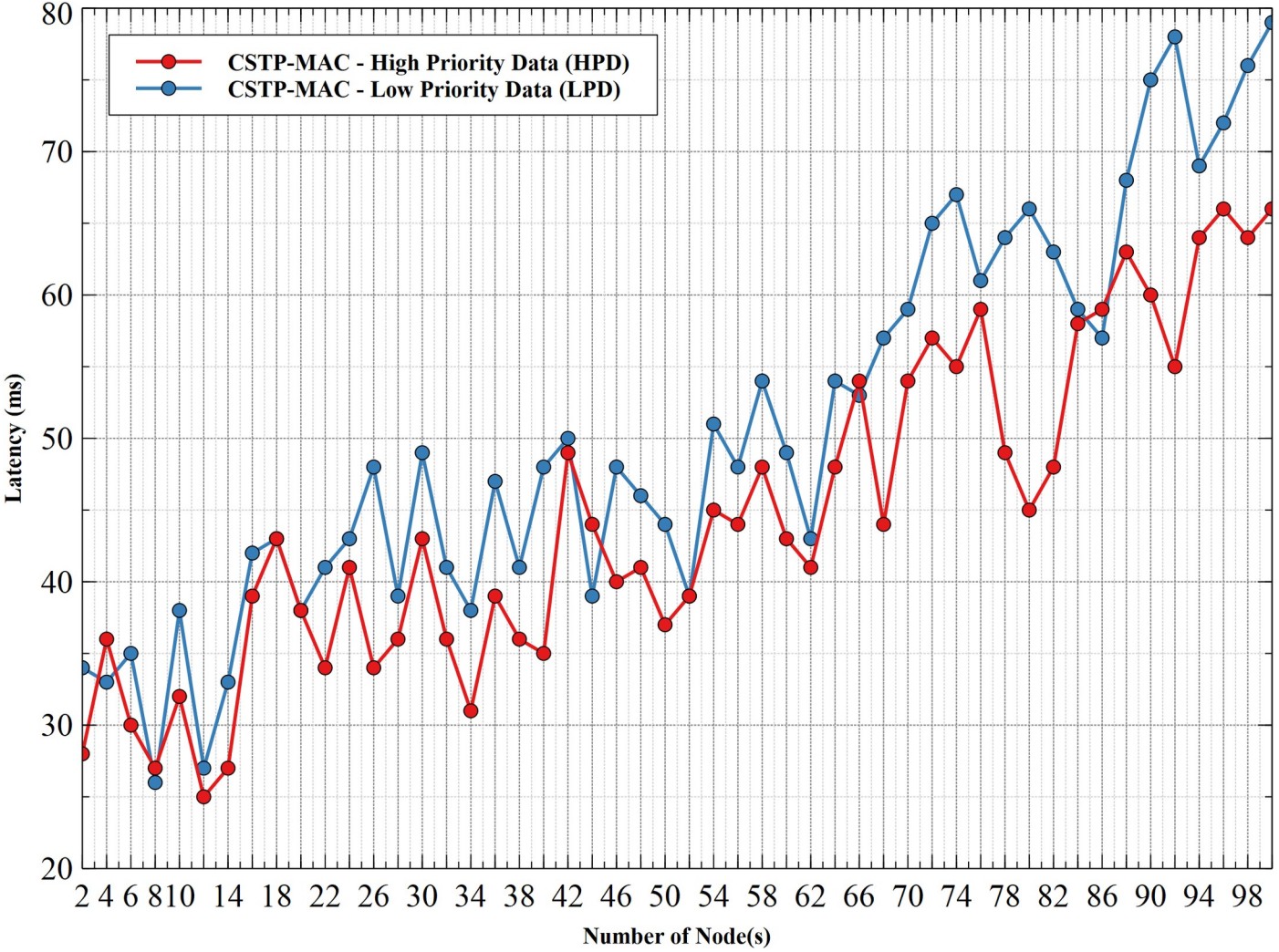

**Fig 12. Latency–HPD vs LPD.**

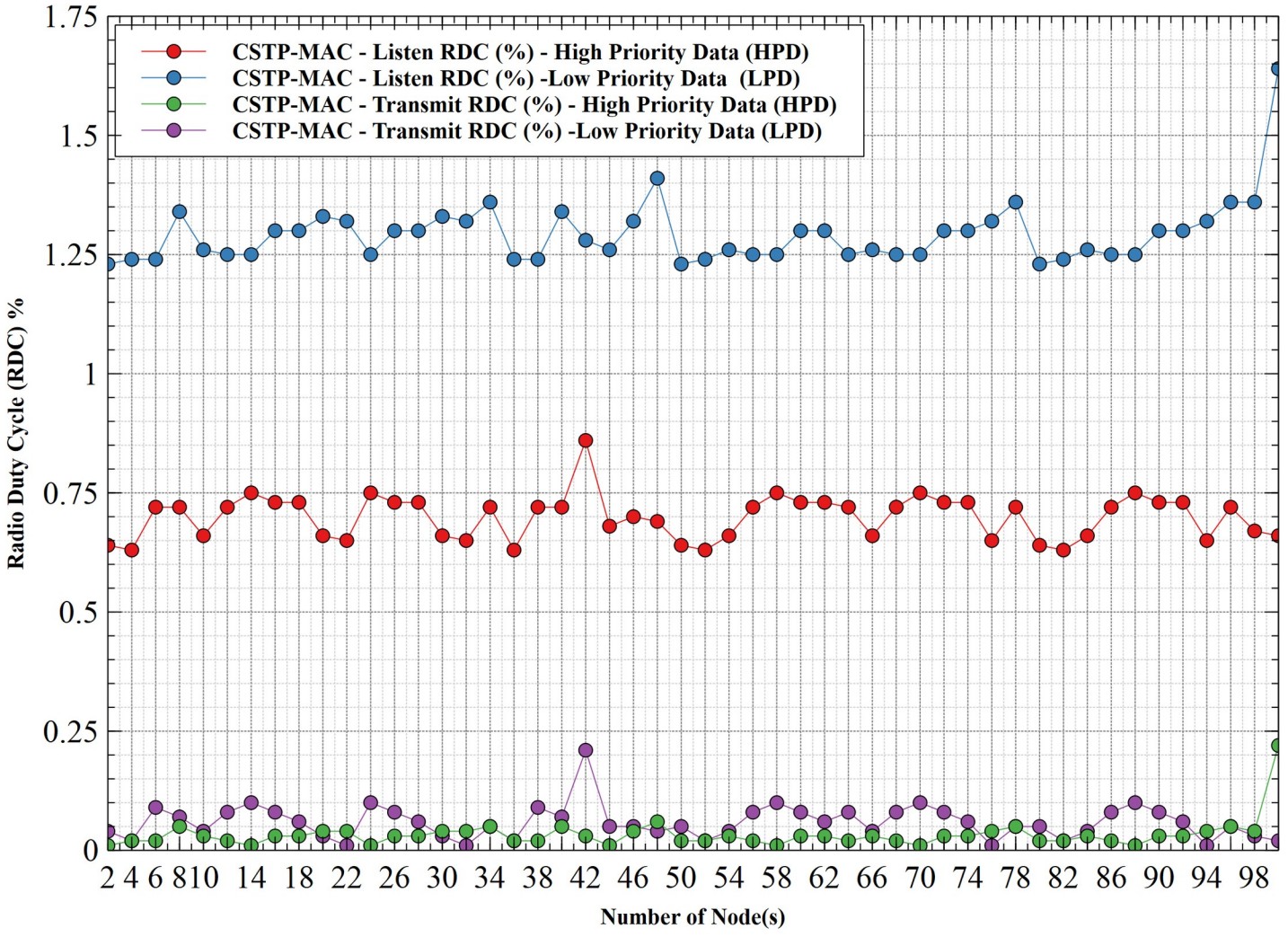

**Fig 13. Radio duty cycle (RDC)–HPD vs LPD.**

operation for both priorities of data. LPD takes a clear lead in the overall duty cycle for both listen and transmit operations. These high duty cycle measures of LPD result from its low throughput and high latency as earlier discussed, which is because of prolonged waiting time experienced during channel contention due to increased collision domain with an increasing number of nodes. Recalling that duty cycle is proportional to power consumption, which also accounts for why LPD nodes consume more operational power compared to HPD nodes.

## 5.2 Benchmarking with priority-based MAC schemes

To understand and appreciate the performance of the proposed scheme compared to existing priority-based MAC implemented schemes, the following subsection details the performance of CSTP-MAC with related schemes [9, 21, 22, 26].

**5.2.1. Packet delivery ratio (PDR).** Fig 14 shows that PG-MAC has the least PDR with its highest measure at 67%. Note that the higher the PDR, the better the network performance. The low PDR measure of PG-MAC is attributed to the fact that all classes of traffic, which is represented by $D_{Type}$, use the same range of BP. This is rather clumsy as there is no unique

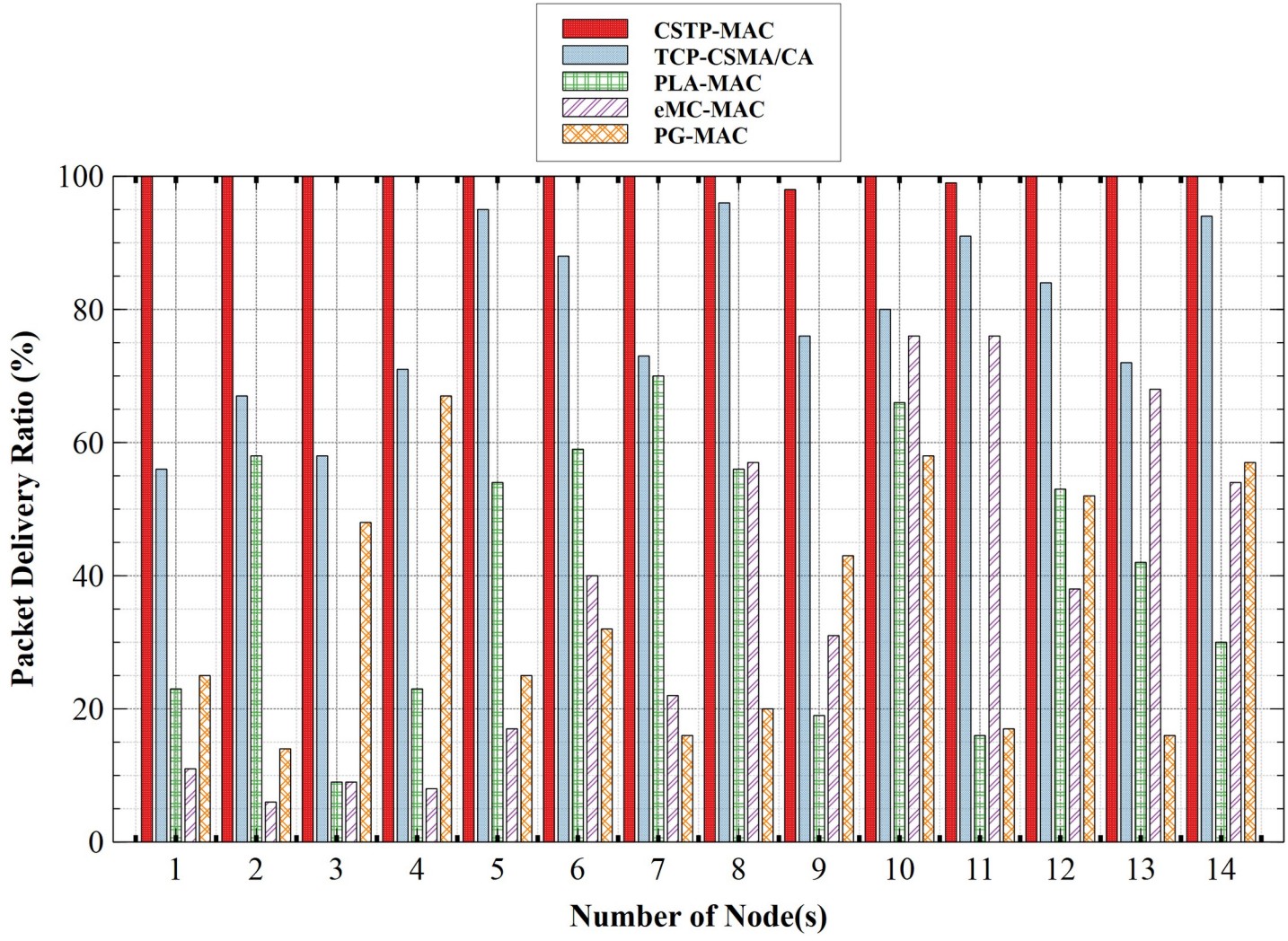

**Fig 14. Packet delivery ratio (PDR).**

distinction on how heterogeneous data can be handled, hence the need for traffic prioritization for this scheme is defeated. This results in a higher data collision rate with increasing network size and reduced throughput. PLA-MAC has its best PDR performance of 70% and 66% at the 7th and 10th nodes, respectively. During channel contention, all classes of traffic use the same BP range resulting in increased packet collision between different traffic classes. In the long run, this affected the PDR which appears to be very unstable. This is closely followed by eMC-MAC with its peak PDR value at 76%. eMC-MAC uses custom Tclass-value to compute back-off slots for nodes of different CS. This exponentially increasing backoff slot leads to increased latency and reduced throughput with a high packet collision rate and reduced PDR, with a peak PDR measure at 96% and followed by 95%. TCP-CSMA/CA allots different classes of service traffic, a unique range of backoff slots to choose from. However, the complexity of the BP algorithm poses high computational overhead on nodes which invariably affects the PDR performance. The proposed MAC scheme, CSTP-MAC, outperforms all previously discussed MAC schemes. CSTP-MAC adopts two traffic classes; HPD with CS = 0 and LPD with CS = 1. This approach greatly reduced the excessive computational complexity experienced by the

TCP-CSMA/CA scheme. CSTP-MAC PDR attained a maximum of 100%, which was fairly maintained with an increasing number of nodes and its minimum value at 98%. For performance comparison, CSTP-MAC had 99% PDR and outperformed its contemporary schemes with 34% where eMC-MAC, PLA-MAC, TCP-CSMA/CA, and PG-MAC attained 13%, 14%, 27%, and 12%, respectively.

**5.2.2. Power consumption.**   The power consumption of all schemes is shown in Fig 15. PLA-MAC is seen to consume the highest power at 19.24mW. While contending to gain access to the shared medium, nodes randomly select their BP from the same backoff slot range, irrespective of their CS, which contributes to increasing power consumption. PG-MAC shows an improvement over PLA-MAC with a maximum value of 16.63mW. PG-MAC assigns a unique BP range to different classes of traffic but remains unchanged with an increasing number of nodes, which adversely led to increased power consumption. Although eMC-MAC shows better power consumption at fewer nodes with peak consumption at 17.43mW, at nodes 13 and 14, its measures exceed that of PLA-MAC and PG-MAC. eMC-MAC provides exponentially increasing backoff range to its nodes, which encourages increased latency with increased

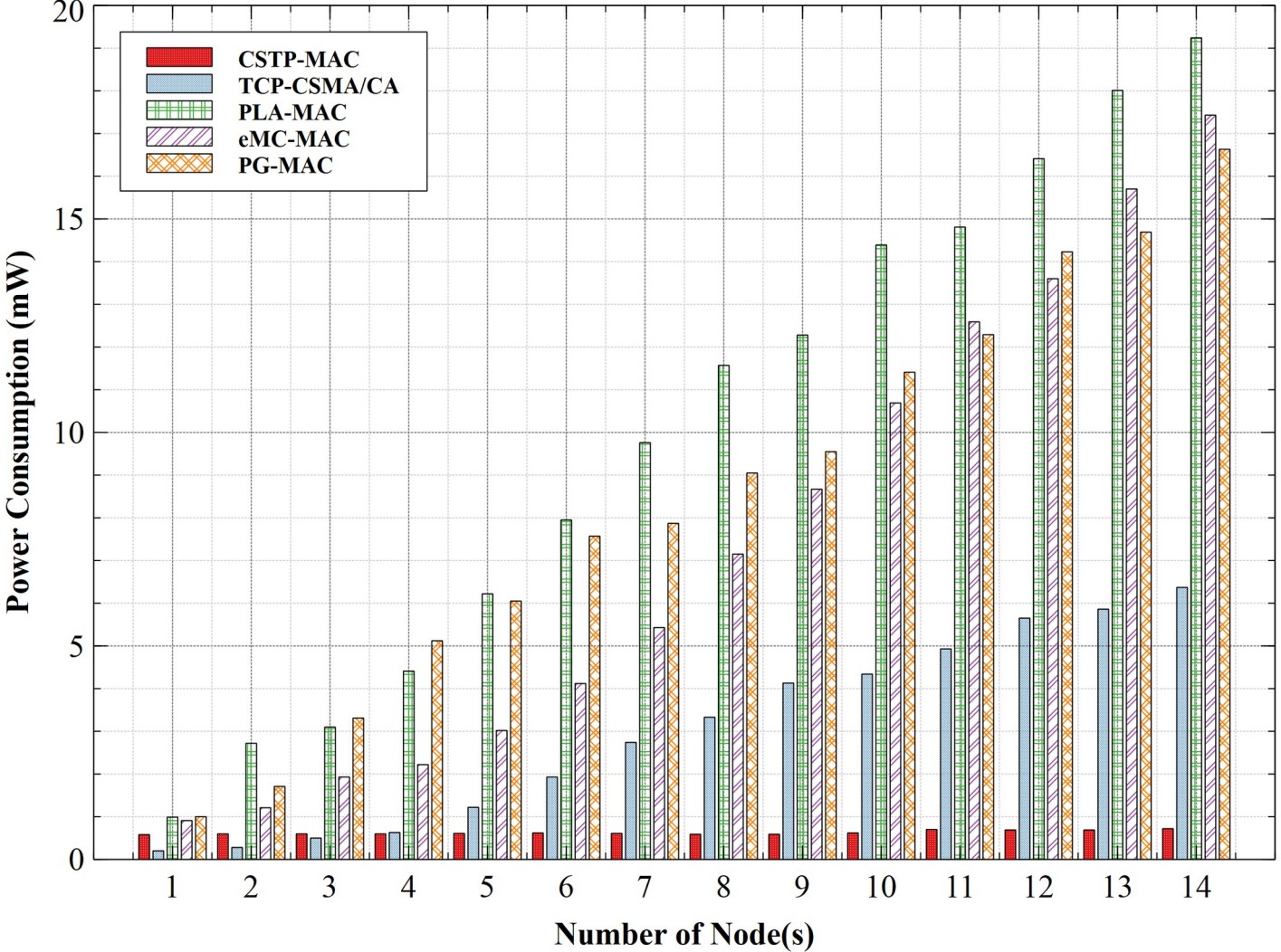

**Fig 15. Power consumption.**

power consumption as revealed at 13 and 14 nodes. TCP-CSMA/CA shows much lower power consumption with a peak value of 6.37mW. However, the computational complexity of this scheme poses overhead and led to increased power consumption. The proposed scheme CSTP-MAC exhibits the most efficient power consumption with drastic reduction (below 1mW), which is maintained with an increased number of nodes and with its highest measure at 0.72mW. The stability of this scheme is attributed to its unique methods of assigning a distinct range of BPs to its different CS data traffic, which changes as the number of nodes increases. For performance comparison, CSTP-MAC uses 2% of power, while TCP-CSMA/CA, PG-MAC, eMC-MAC, and PLA-MAC schemes used 10%, 29%, 25%, and 33%, respectively. CSTP-MAC outperformed its contemporary schemes by a power-saving measure of 98%, which translates to a longer operational lifetime.

**5.2.3. Throughput.** Fig 16 shows the throughput across all schemes. PLA-MAC shows a fairly consistent throughput with increasing node number, but at higher node number, its value starts to drop compared to that of its contemporary schemes. This occurs because the scheme adopts the same range of BPs across all class of service data traffic, thus this approach

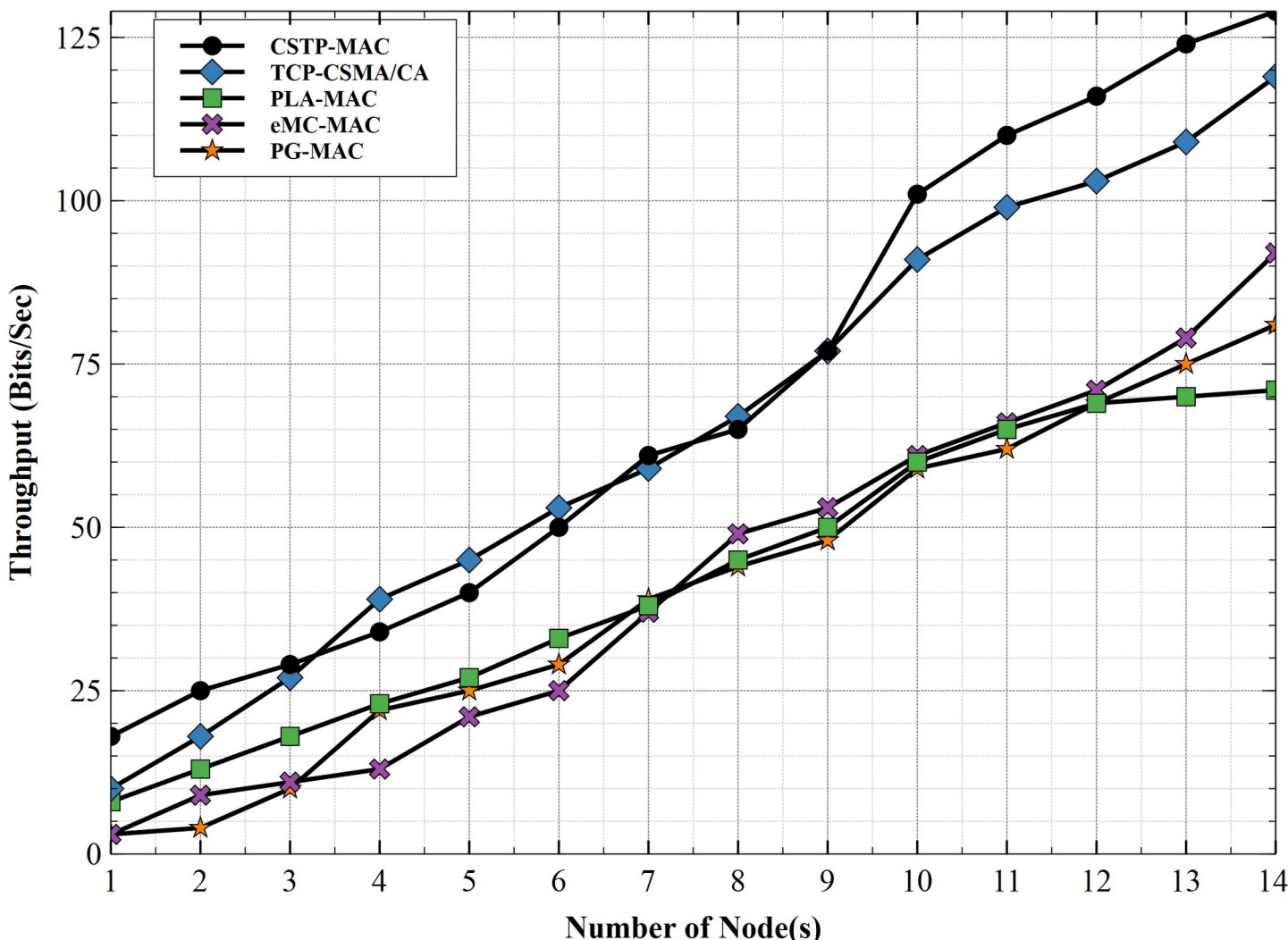

**Fig 16. Throughout.**

causes high priority traffic class to be treated the same as low priority traffic class. This adversely impacts the throughput as node number increases. eMC-MAC performs slightly better than PLA-MAC. The exponentially increasing range of backoff slots made available for different CS traffic remains unchanged with an increasing number of nodes, which encourages packets collision with an adverse effect on network throughput. PG-MAC used the same backoff slot range with all traffic class even with an increasing number of nodes. This practice inhibits the improvement of throughput even though it increases fairly. TCP-CSMA/CA shows better throughput performance. The traffic classification method adopted by this scheme impacts positively on its throughput performance where different CS traffic is treated with unique BPs. The proposed scheme, CSTP-MAC, exhibits better throughput performance than all previously discussed schemes, especially at a higher number of nodes, particularly from the 9th node. This improved performance is attributed to how different traffic classes are made to choose BPs from uniquely different range of backoff which changes with an increasing number of nodes. This approach positively impacts the overall performance of the network. For performance comparison, CSTP-MAC attains 27%, while its contemporary schemes, TCP-CSMA/CA, PG-MAC, eMC-MAC, and PLA-MAC, attained 26%, 16%, 16%, and 16%, respectively. CSTP-MAC shows better throughput performance.

**5.2.4. Latency.** Fig 17 shows the latency of all MAC schemes considered. PG-MAC shows the highest latency at 597ms, which steadily increases with an increasing number of nodes. In this scheme, all traffic classes are treated similarly even with an increasing number of nodes as nodes choose BP from the same range of time-slots. This approach adversely impacts the latency performance as node number increases. PLA-MAC shows a fairly improved latency at 580ms, as compared to PG-MAC. While using exponentially increasing backoff range, this backoff slot range remains the same as node number increases. This approach encourages packet collision with increased latency. eMC-MAC shows better latency over the previously discussed schemes with a peak value at 424ms. Similar to PLA-MAC with an exponentially increasing range of BPs, different traffic classes are treated the same without distinction. TCP-CSMA/CA scheme shows a much-improved latency with its highest value at 294ms, as the scheme assigns different BP range to different traffic priorities. This implementation contributes to the improved latency of the scheme. The proposed scheme, CSTP-MAC, exhibits the most improved latency of all schemes considered, which is due to the well-implemented CS for different priorities of data traffic. Traffics are given unique backoff slots that change with increasing node number, hence access to the shared channel/medium is granted based on the priority of data traffic with little or no contention. This approach improves the latency greatly with a peak value of 241ms and hence shows that CSTP-MAC outperforms its contemporary schemes.

## 6 Conclusions, limitation and future work

This study presents an improvement of the standard IEEE 802.15.4 CSMA/CA scheme with consideration for data prioritization of heterogeneous data generated by WSN. The approaches of channel contention scheme proposed in the literature for prioritized data have been insufficient to address the heterogeneous nature of WSN with varying data priorities. In this article, the CSTP-MAC scheme has been proposed to proffer a solution to this issue. The scheme adopts the concept of CS to implement data prioritization by classifying heterogeneous data into HPD and LPD with priority tags CS 0 and 1, respectively. The traditional MAC frame format of the IEEE 802.15.4 was modified to accommodate data prioritization, with improved beacon called *Beacon Plus Plus (B$^{++}$)*. Subsequently, the traditional binary exponential backoff algorithm was also modified to obtain unique backoff expressions peculiar to the

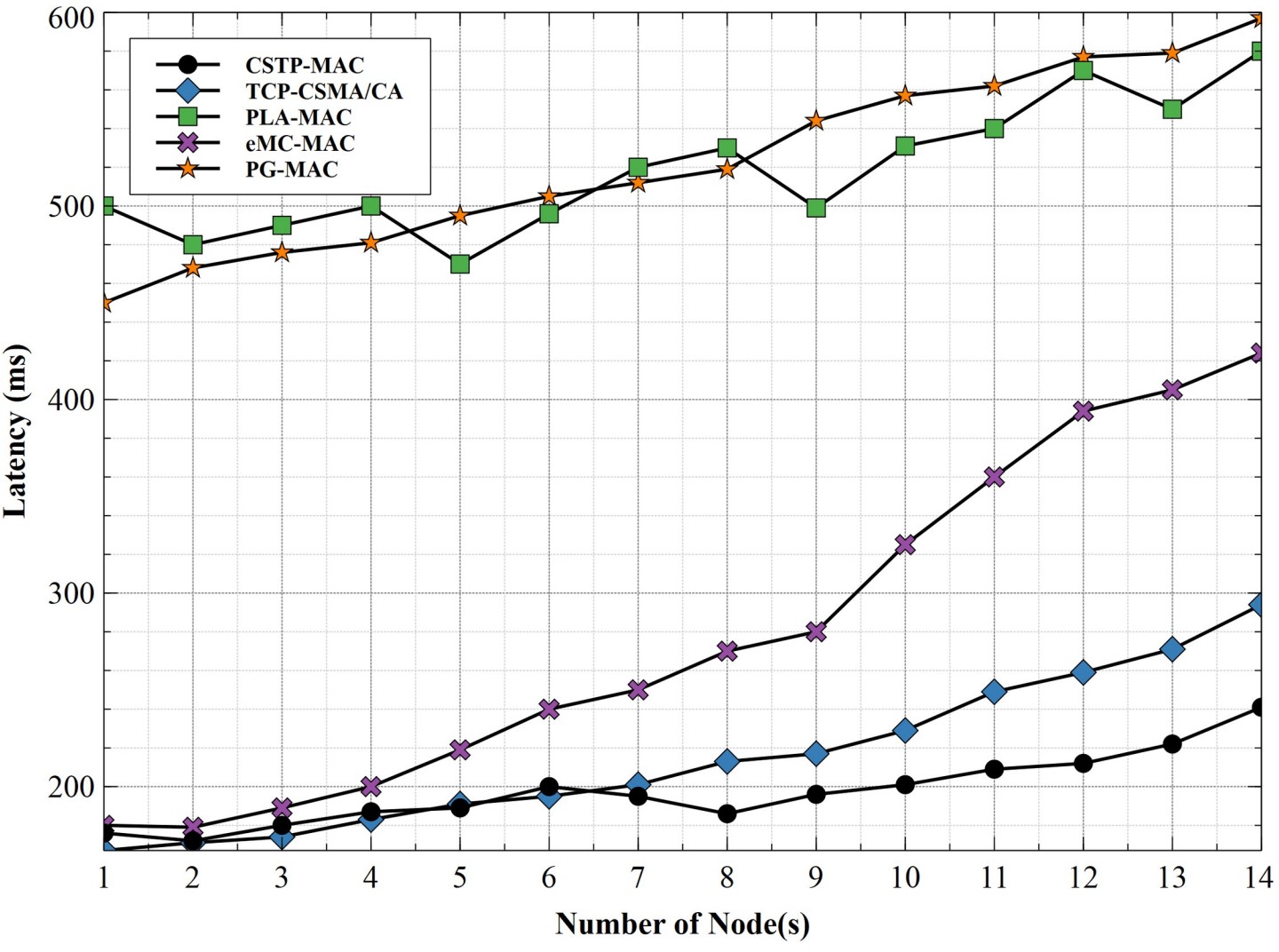

**Fig 17. Latency.**

data priority status. This implementation achieved remarkable improvement in how nodes gain access to the shared medium of transmission while making provision for data of varying heterogeneity and priorities. Validating the proposed scheme, results show that HPD performed better than LPD by gaining quicker access to the shared transmission medium with improved performance compared to LPD, which is shown by the network performance metric statistics. Subsequently, CSTP-MAC outperformed other related priority-based MAC schemes in regards to PDR, throughput, power consumption, and latency. This improvement translates to a prolonged operational lifetime for WSN. The limitation of this work is that it only addresses data heterogeneity and priority at the data link layer. Our future work will focus on data prioritization implementation at the network layer of the WSN protocol stack.

## Supporting information

**S1 File. This ZIP file contains a dataset in Excel format (XLSX) used to plot Figs 6, 9 to 17.**
(ZIP)

## Acknowledgments

The research would not have been thoroughly completed without the support provided by University Teknologi Malaysia (UTM). The lab and academic resources have been very helpful. Thank you so much.

## Author Contributions

**Conceptualization:** Innocent Uzougbo Onwuegbuzie.

**Data curation:** Innocent Uzougbo Onwuegbuzie.

**Formal analysis:** Innocent Uzougbo Onwuegbuzie.

**Investigation:** Innocent Uzougbo Onwuegbuzie, Shukor Abd Razak, Ismail Fauzi Isnin, Tasneem S. J. Darwish, Arafat Al-dhaqm.

**Methodology:** Innocent Uzougbo Onwuegbuzie, Tasneem S. J. Darwish, Arafat Al-dhaqm.

**Project administration:** Innocent Uzougbo Onwuegbuzie, Shukor Abd Razak, Ismail Fauzi Isnin.

**Resources:** Innocent Uzougbo Onwuegbuzie, Shukor Abd Razak, Ismail Fauzi Isnin.

**Software:** Innocent Uzougbo Onwuegbuzie, Shukor Abd Razak, Ismail Fauzi Isnin.

**Supervision:** Shukor Abd Razak, Ismail Fauzi Isnin, Tasneem S. J. Darwish, Arafat Al-dhaqm.

**Validation:** Innocent Uzougbo Onwuegbuzie, Shukor Abd Razak, Ismail Fauzi Isnin, Tasneem S. J. Darwish, Arafat Al-dhaqm.

**Visualization:** Innocent Uzougbo Onwuegbuzie.

**Writing – original draft:** Innocent Uzougbo Onwuegbuzie.

**Writing – review & editing:** Innocent Uzougbo Onwuegbuzie, Tasneem S. J. Darwish, Arafat Al-dhaqm.

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
