## [Decision Letter · Decision Letter 0]

5 May 2020

PONE-D-20-07077

Optimized Backoff Scheme for Prioritized Data in Wireless Sensor Networks: A Class of Service Approach

PLOS ONE

Dear Mr. Onwuegbuzie,

Thank you for submitting your manuscript to PLOS ONE. After careful consideration, we feel that it has merit but does not fully meet PLOS ONE’s publication criteria as it currently stands. Therefore, we invite you to submit a revised version of the manuscript that addresses the points raised during the review process.

We would appreciate receiving your revised manuscript by Jun 19 2020 11:59PM. To enhance the reproducibility of your results, we recommend that if applicable you deposit your laboratory protocols in protocols.io, where a protocol can be assigned its own identifier (DOI) such that it can be cited independently in the future. For instructions see: http://journals.plos.org/plosone/s/submission-guidelines#loc-laboratory-protocols

We look forward to receiving your revised manuscript.

Kind regards,

Chi-Tsun Cheng, Ph.D., M.Sc., B.Eng.

Academic Editor

PLOS ONE

Journal Requirements:

3. Please include a copy of Table 5 which you refer to in your text on line 367.

4. We note you have included a table to which you do not refer in the text of your manuscript. Please ensure that you refer to Table 4 in your text; if accepted, production will need this reference to link the reader to the Table.

Additional Editor Comments (if provided):

A new back-off scheme has been proposed for IEEE 802.15.4. Simulation results show that the proposed scheme can perform better than other existing variants. However, the novelty of the proposed scheme has not been fully elaborated in the paper. The application scenarios and the corresponding limitations of the proposed scheme need to be discussed further. For the analysis, the authors should also explain further on what makes the proposed scheme performs better than the others. The paper is therefore recommended to receive a major revision.

Reviewers' comments:

Reviewer's Responses to Questions

**Comments to the Author**

1. Is the manuscript technically sound, and do the data support the conclusions?

Reviewer #1: Partly

Reviewer #2: Partly

Reviewer #3: Yes

Reviewer #4: Yes

2. Has the statistical analysis been performed appropriately and rigorously? 

Reviewer #1: I Don't Know

Reviewer #2: N/A

Reviewer #3: Yes

Reviewer #4: Yes

3. Have the authors made all data underlying the findings in their manuscript fully available?

Reviewer #1: Yes

Reviewer #2: Yes

Reviewer #3: Yes

Reviewer #4: Yes

4. Is the manuscript presented in an intelligible fashion and written in standard English?

Reviewer #1: No

Reviewer #2: Yes

Reviewer #3: Yes

Reviewer #4: Yes

5. Review Comments to the Author

Reviewer #1: The paper deals with CoSTP-MAC, a custom version of backoff scheme for prioritized data in IEEE 802.15.4 Wireless Sensor Networks. The paper describes the proposed approach and provides a simulative assessment of CoSTP-MAC with respect other solutions in the literature.

Several issues should be solved.

My comments are as follows:

• Examples of the possible use of the proposed approach in real application should be provided. Please provide references.

• “The protocol that controls and regulates the functioning of WSN is the IEEE 802.15.4”. Other technologies were considered in the literature to implement WSNs. Please clear this concept.

• The Priority Channel Access (PCA) mechanism provided by IEEE 802.15.4 should be discussed. Please provide references about this mechanism.

• Please introduce the concepts of Superframe duration and Beacon Interval before Fig. 1.

• “In the literature, some existing data prioritization schemes has attempted to proffer solution to this issue”. Please provide references.

• The paper motivation should be explained in a better way.

• In the text both CoS and CS are used for Class of Services. Please correct.

• Are multi-hop communications supported by the proposed approach?

• The quality of Fig. 6 should be improved.

• The simulated scenario should be described in a better way. Is it a realistic scenario? Please provide references. What propagation model was used?

• Please define the Packet Delivery Ratio (PDR) using an equation.

• The definition of Latency (End-to-end Delay) is not very clear. It refers to End-to-end delay or to Round Trip Time (RTT)? Please clear this concept.

• The obtained results should be discussed in a better way. For instance:

- Please discuss about the results presented in Fig. 7, e.g., when the number of nodes is equal to 26.

- The results in Fig. 8.

- Please discuss about the results presented in Fig. 9, e.g., when the number of nodes is equal to 25.

• Can the proposed approach be easily implemented on COTS devices? An implementation of the proposed approach should be important to demonstrate the feasibility of CoSTP-MAC on real devices.

• Are you able to discuss possible future work linked to this paper?

• There are some language flaws and typos (e.g., “alorithm”, “possble”, “perculiar”, “perfromed”, etc). Please proofread the paper before final submission.

Reviewer #2: In this paper, the authors present the “Optimized backoff scheme for prioritized data in wireless sensor networks: a class of service approach”. The specific topic is very interesting, as the Internet of things (IoTs) and healthcare applications have recently attracted a lot of attention.

I have the following comments to improve the paper.

1) The novelty and the contribution of the paper are questionable. There are many related works (most of them are not cited in the paper) that study MAC issues and IEEE 802.15.4 standards, and it is not clear how this work differentiates from them and goes beyond the state of the art.

2) The authors focus on the IEEE 802.15.4 Standard, although there are other standards which are designed for specific applications such as IEEE 802.15.6 Standard, which is designed especially for WBANs.

3) The technical depth of the paper is not sufficient. There is no analysis, while the proposed protocol is a simple variation of existing methods.

4) The figure of CoSTP-MAC flowchart for slotted CSMA/CA is not clear, please replace with a high-quality figure, and add the description.

5) The scenario is far from the real scenario where WSN will have to operate. The authors should conduct a literature review focusing on the operating conditions of WSN.

6) In the performance comparisons between high priority data (HPD) and low priority data (LPD), in some cases, the performance of LPD is better than the HPD (especially in Fig.7 when the number of nodes is 28, and in Fig. 9). Please provide the appropriate description.

7) It would be a good idea that the authors can elaborate on how the performance would be with respect to the number of devices increases not limited up to 30. Please compare the performance of the proposed scheme by increasing the number of nodes (Fig.12- Fig. 15)

Reviewer #3: This study presents an improvement of the standard IEEE 802.15.4 CSMA/CA algorithm with consideration for data prioritization of heterogeneous data generated by WSN. The proposed CoSTP-MAC scheme implements data prioritization by classifying heterogeneous data into HPD and LPD with priority tags; CS 0 and 1 respectively. The traditional MAC frame format of the IEEE 802.15.4 was modified to accommodate data prioritization, with improved beacon called Beacon Plus Plus (B++). Subsequently, the traditional binary exponential backoff algorithm was also modified to obtain unique backoff algorithms peculiar to the data priority status. This implementation achieved fairness and improvement on how nodes gain access to the shared medium of transmission with respect to their priority status. Validating the proposed scheme, results show that HPD performed better than LPD by gaining quicker access to the shared transmission medium with improved performance compared to LPD. Subsequently, CoSTP-MAC outperformed other schemes with respect to PDR, throughput, latency, radio duty cycle and power consumption with improved operational longevity.

Overall, the paper presents a reasonable scientific contribution. However, the reviewer has few comments given below.

① Authors must review the acronyms and their extension (on the first appearance). Apart from abstract, the full name of the abbreviations should be provided for the first-time appearance, and then only the abbreviations should be used, “Backoff Period (BP)” is defined several times. Moreover, there is no synchronization in the abbreviations and their extensions; some appear as “Personal Area Network (PAN)”, while others appear as “class of service (CoS)”. Also, please provide the full form of “MAC”, “CoSTP-MAC”, etc. at its first appearance.

② Line#119, “Clear Channel Access (CCA)” should be replaced by “Clear Channel Assessment (CCA)”

③ The undefined terms in some of the equations should be defined anywhere in the section.

④ Please proofread the paper again to correct the typographical errors.

⑤ Also, there is lot of room for language improvement.

⑥ Most of the references are up-to-date, but some older references can be replaced with the related and latest ones.

⑦ Try to include better quality picture for figure 6.

⑧ To generate more research interest, the authors should include some future works in the article.

The suggested revisions must be incorporated in the revised manuscript.

Reviewer #4: Good effort made in IEEE 802.15.4 mac. The authors could revise the below points:

1. CoSTP-MAC acronym expansion needed on first usage in the abstract section. Also,Under what scenario/configuration/topology/mac-mode was it possible to achieve 99% PDR and 98% power saving capability needs to provided with clarity in abstract section?

2. The initial lines of Introduction section has grammatical errors. Bulk-citing of references in introduction section to properly correlate. Reference no 8 - paper title to be expanded.

3. Eqn 1 - error in power raised "-1" term [0, 2^BE - 1].

4. Fig 2B) typo - Black nodes are RT nodes

5. Section 3.1.1 typos - known as BEB, BE=3 {0,1,2..

6. Section 3.2 - line 274-276 are ambiguous to convey the meaning.

the relation in arriving the equations for the backoff limits could be presented in a more clearer manner. The modeling correctness could be proved! What would happen when more nodes closer to each other with same HPD priority contends with these backoff relation? Will it lead to increased contention and collison with modified backoff limit?

7. In section 4.1,

Justify the reason for selection of number of nodes for the simulation and impact of scaling more nodes in single-hop?

Why CBR is used as the traffic model? Are HPD nodes scheduled at random times to send the traffic during the simulation?

Will this work on multi-hop networks?

6. PLOS authors have the option to publish the peer review history of their article (what does this mean?). If published, this will include your full peer review and any attached files.

Reviewer #1: No

Reviewer #2: No

Reviewer #3: No

Reviewer #4: No

---

## [Author Response · Author response to Decision Letter 0]

22 Jun 2020

Editors' Comments and Responses by the Authors

1. Please include a copy of Table 5 which you refer to in your text on line 367 

Response

Thank you for pointing this out.

The actual Table meant to be referenced is Table 4 and not Table 5. The correction has been made. Table 5 is now changed to Table 4. See subsection 4.1

2. We note you have included a table to which you do not refer in the text of your manuscript. Please ensure that you

refer to Table 4 in your text; if accepted, production will need this reference to link the reader to the Table 

Response

Table 4 has been adequately referenced in the paper. See subsection 4.1

3. A new back-off scheme has been proposed for IEEE 802.15.4. Simulation results show that the proposed scheme can perform better than other existing variants. However, the novelty of the proposed scheme has not been fully elaborated in the paper. The application scenarios and the corresponding limitations of the proposed scheme need to be discussed further. For the analysis, the authors should also explain further on what makes the proposed scheme performs better than the others. 

Response

Thank you for pointing this out.

The novelty of the CSTP-MAC scheme has been elaborated in Section 3. The scheme proposed an improvement on the IEEE 802.15.4 standard with consideration for heterogeneous data (real-time and non-real-time). The scheme computes a range of backoff periods (BP) with expressions peculiar to the priority and class of the data. The proposed scheme shows improved network performance as compared to the Standard IEEE 802.15.4 scheme and other related schemes. See section 5 of Results and Discussions. 

The limitation of the scheme is that it only addresses data prioritization at the data link layer. This serves as motivation for our future work, which is to proffer similar improvement for the network layer of the WSN protocol stack. This is discussed in Section 6 of Conclusion, limitation, and future work.

To show the support for the scalability of the scheme, the number of nodes was increased from 30 to 100, and analysis was performed based on performance metrics. Results show that HPD (real-time data) has better network performance than LPD (non-real-time data), subsequently, the CSTP-MAC scheme also has better network performance than related schemes. See sections 4 and 5

Reviewer 1 Comments and Responses by the Authors

1. Examples of the possible use of the proposed approach in real application should be provided. Please provide references. 

Response

Thank you for pointing this out.

Example of the possible use of the proposed approach has been provided in section 3. The CSTP-MAC scheme applies to Biomedical Sensors in Wireless Body Area Network (WBAN) in the field of medicine as well as in general WSN applications such as home, industry, logistics, military, where data priority plays a major role.

References are provided. See section 3

2. “The protocol that controls and regulates the functioning of WSN is the IEEE 802.15.4”. Other technologies were considered in the literature to implement WSNs. Please clear this concept. 

Response

Thank you for pointing this out.

The statement has been clarified and presented in a better perspective. See section 1 of Introduction.

3. The Priority Channel Access (PCA) mechanism provided by IEEE 802.15.4 should be discussed. Please provide references about this mechanism. 

Response

The IEEE 802.15.4 standard does not provide Priority Channel Access (PCA) to nodes except for beacon transmission which is given priority access to the channel whenever necessary. This is discussed in section 2 and references have been provided.

4. Please introduce the concepts of Superframe duration and Beacon Interval before Fig. 1. Thank you for pointing this out. 

Response

The concept of superframe duration and Beacon Interval has been introduced in subsection 1.1

5. “In the literature, some existing data prioritization schemes has attempted to proffer solution to this issue”. Please provide references. 

Response

References have been provided for this statement in subsection 1.1

6. The paper motivation should be explained in a better way. 

Response

Thank you for pointing this out.

The Paper motivation has been improved. See section 2.

7. In the text both CoS and CS are used for Class of Services. Please correct. 

Response

All CoS has been changed to CS. That is CoSTP-MAC is now presented as CSTP-MAC

8. Are multi-hop communications supported by the proposed approach? 

Response

Yes, the CSTP-MAC scheme supports both single-hop and multi-hop communications. This is explained in section 3 and detailed by Fig 3

9. The quality of Fig. 6 should be improved 

Response

Thank you for pointing this out. 

The figure is no longer represented by Fig 6 but by Figs 7 and 8 with improved quality. See subsection 3.3

10. The simulated scenario should be described in a better way. Is it a realistic scenario? Please provide references. What propagation model was used? 

Response

Thank you for pointing this out. 

The simulated scenario is realistic and has been described in a better way in section 4.0 and subsection 4.1. The adopted propagation model is Unit Disk Graph Medium (UDGM): Distance Loss, this is shown in Table 4

11. Please define the Packet Delivery Ratio (PDR) using an equation. 

Response

Packet Delivery Ratio has been defined by an equation in subsection 4.2 

12. The definition of Latency (End-to-end Delay) is not very clear. It refers to End-to-end delay or to Round Trip Time (RTT)? Please clear this concept. 

Response

The definition of Latency has been clarified in subsection 4.2

13. The obtained results should be discussed in a better way. For instance:

- Please discuss about the results presented in Fig. 7, e.g., when the number of nodes is equal to 26.

- The results in Fig. 8.

- Please discuss about the results presented in Fig. 9, e.g., when the number of nodes is equal to 25. 

Response

Thank you for pointing this out. 

Figs 7 – 11 are now represented as Figs 9 – 13. The number of nodes used for simulation to obtain Figs 9 – 13 was increased to 100 and the simulation reperformed. A better explanation has been offered for the newly obtained Figs.

14. Can the proposed approach be easily implemented on COTS devices? An implementation of the proposed approach should be important to demonstrate the feasibility of CoSTP-MAC on real devices. 

Response

Yes, the proposed approach can be easily implemented on a real-life sensor device. 

The emulation of Zolertia Z1 mote incorporated in the simulator (Contiki OS) which has a real-life physical mote was used as the nodes for simulation. Z1 mote runs on the MPS430 microprocessor with the CC2420 RF transceiver. See Table 4 for details.

15. Are you able to discuss possible future work linked to this paper? 

Response

Future work is discussed in section 6 of Conclusions, limitation and future work.

16. There are some language flaws and typos (e.g., “alorithm”, “possble”, “perculiar”, “perfromed”, etc). Please proofread the paper before final submission. 

Response

Thank you for pointing this out. 

The language flaws and typographical errors have been corrected. The paper has been proofread.

Reviewer 2 Comments and Responses by the Authors

1. The novelty and the contribution of the paper are questionable. There are many related works (most of them are not cited in the paper) that study MAC issues and IEEE 802.15.4 standards, and it is not clear how this work differentiates from them and goes beyond the state of the art. 

Responses

The novelty and contribution of this paper are presented in section 3. More recent papers have been cited and the contribution of this work with respect to the IEEE 802.15.4 standard is presented in section 3.

2. The authors focus on the IEEE 802.15.4 Standard, although there are other standards which are designed for specific applications such as IEEE 802.15.6 Standard, which is designed especially for WBANs. 

Response

The goal of the authors is not just on a standard that focusses on WBAN but on a versatile standard that cuts across other WSN applications. See section 3

3. The technical depth of the paper is not sufficient. There is no analysis, while the proposed protocol is a simple variation of existing methods. 

Response

The technical depth has been improved with detailed analysis. See section 3

4. The figure of CoSTP-MAC flowchart for slotted CSMA/CA is not clear, please replace with a high-quality figure, and add the description. 

Response

Thank you for pointing this out.

The scheme is now known as CSTP-MAC, the figure of CSTP-MAC flowchart is now presented by Figs 7 and 8 with improved quality. See subsection 3.3

5. The scenario is far from the real scenario where WSN will have to operate. The authors should conduct a literature review focusing on the operating conditions of WSN. 

Response

Thank you for pointing this out.

The scenario is improved upon and now presented in a more realistic manner, this is detailed in section 4

6. In the performance comparisons between high priority data (HPD) and low priority data (LPD), in some cases, the performance of LPD is better than the HPD (especially in Fig.7 when the number of nodes is 28, and in Fig. 9). Please provide the appropriate description. 

Response

Thank you for pointing this out.

The simulation for performance comparison between HPD and LPD was reperformed with an increased number of nodes from 30 to 100 and the results graphically presented from Figs 9 – 13. An appropriate description has also been provided for this scenario in subsection 5.1 

7. It would be a good idea that the authors can elaborate on how the performance would be with respect to the number of devices increases not limited up to 30. Please compare the performance of the proposed scheme by increasing the number of nodes (Fig.12- Fig. 15) 

Response

Thank you for your suggestion.

The number of nodes was increased from 30 to 100 and improved description has been provided in section 5 of Results and discussions

Reviewer 3 Comments and Responses by the Authors

1. Authors must review the acronyms and their extension (on the first appearance). Apart from abstract, the full name of the abbreviations should be provided for the first-time appearance, and then only the abbreviations should be used, “Backoff Period (BP)” is defined several times. Moreover, there is no synchronization in the abbreviations and their extensions; some appear as “Personal Area Network (PAN)”, while others appear as “class of service (CoS)”. Also, please provide the full form of “MAC”, “CoSTP-MAC”, etc. at its first appearance. 

Response

Thank you for pointing this out.

All acronym-based terms have been restricted to full usage on the first time, they are represented by their respective acronym in subsequent occurrence

The full form of CSTP-MAC is now presented in the Abstract.

2. Line#119, “Clear Channel Access (CCA)” should be replaced by “Clear Channel Assessment (CCA)” 

Response

Thank you for pointing this out.

Clear Channel Access has been replaced by Clear Channel Assessment. See subsection 2.1

3. The undefined terms in some of the equations should be defined anywhere in the section. 

Response

Thank you for pointing this out.

The undefined terms in some of the equations have been defined.

4. Please proofread the paper again to correct the typographical errors. 

Response

Thank you for pointing this out.

The paper has been proofread and typographical errors corrected.

5. Also, there is lot of room for language improvement. 

Response

Thank you for pointing this out.

The language has been improved upon

6. Most of the references are up-to-date, but some older references can be replaced with the related and latest ones. 

Response

Thank you for your suggestion.

The older references have been replaced with related and recent ones.

7. Try to include better quality picture for figure 6. 

Response

Thank you for pointing this out.

Fig 6 has been replaced with Figs 7 and 8 with improved picture quality. See section 3.3

8. To generate more research interest, the authors should include some future works in the article. 

Response

Thank you for your suggestion.

Future work has been included in section 6 of Conclusions, limitation and future work

Reviewer 4 Comments and Responses by the Authors

1. CoSTP-MAC acronym expansion needed on first usage in the abstract section. Also, under what scenario/configuration/topology/mac-mode was it possible to achieve 99% PDR and 98% power saving capability needs to provide with clarity in abstract section? 

Response

Thank you for pointing this out.

The CoSTP-MAC has been replaced with CSTP-MAC and the acronym has been expanded on the first usage in the Abstract.

The scenario/configuration/topology/MAC-mode under which 99% PDR was attained in described in section 4, subsequently, this has been updated in the Abstract.

2. The initial lines of Introduction section have grammatical errors. Bulk citing of references in introduction section to properly correlate. Reference no 8 - paper title to be expanded. 

Response

Thank you for pointing this out.

The grammatical errors in the initial lines of the Introduction have been corrected and the bulk citation has been addressed.

The full title of reference number 8 has been expanded in the Reference section.

3. Eqn 1 - error in power raised "-1" term [0, 2^BE - 1]. 

Response

Thank you for pointing this out.

The error of power raised (exponent) in the equation has been corrected and is now represented as Eq 4.

4. Fig 2B) typo - Black nodes are RT nodes 

Response

Thank you for pointing this out.

The correction has been made and the figure is updated and represented by Fig 3

5. Section 3.1.1 typos - known as BEB, BE=3 {0,1,2.. 

Response

Thank you for pointing this out.

The correction has been affected. See subsection 3.1.1

6. Section 3.2 - line 274-276 are ambiguous to convey the meaning.

The relation in arriving the equations for the backoff limits could be presented in a clearer manner. The modeling correctness could be proved! What would happen when more nodes closer to each other with same HPD priority contends with these backoff relation? Will it lead to increased contention and collision with modified backoff limit? 

Response

Thank you for pointing this out.

The ambiguity is clarified. The statement is represented in a clearer perspective.

Fig 3b depicts a scenario of nodes with similar priority class closer to each other as High Priority Node (HPNs) and Low Priority Nodes (LPNs). Access to the shared transmission medium is based on first come first serve. The modified backoff limits reduced the contention and collision as nodes select nonconflicting backoff time from the range of controlled backoff period. See section 5 of Results and discussions

7. In section 4.1, justify the reason for selection of number of nodes for the simulation and impact of scaling more nodes in single-hop? 

• Why CBR is used as the traffic model? Are HPD nodes scheduled at random times to send the traffic during the simulation?

• Will this work on multi-hop networks? 

Response

Thank you for pointing this out.

To show the support for the scalability of the CSTP-MAC scheme, the number of nodes was increased from 30 to 100 and the simulation reperformed. The results are discussed in section 5

Constant Bit Rate (CBR) is used as data is generated at a constant rate, subsequently, the resource-constrained nature of sensor nodes/devices is favored by CBR. This is detailed in section 4

For the simulation, all node including HPNs are scheduled to send traffic at a constant rate, hence the use of CBR

Yes, the CSTP-MAC scheme supports both the single-hop and multi-hop propagation model. See sections 3 and 4

---

## [Decision Letter · Decision Letter 1]

22 Jul 2020

Optimized backoff scheme for prioritized data in wireless sensor networks: a class of service approach

PONE-D-20-07077R1

Dear Dr. Onwuegbuzie,

We’re pleased to inform you that your manuscript has been judged scientifically suitable for publication and will be formally accepted for publication once it meets all outstanding technical requirements.

Kind regards,

Chi-Tsun Cheng, Ph.D., M.Sc., B.Eng.

Academic Editor

PLOS ONE

Additional Editor Comments (optional):

All issues in the last round of review have been addressed properly in the current verison. The paper has merits and is recommended to be accepted.

Reviewers' comments:

Reviewer's Responses to Questions

**Comments to the Author**

1. If the authors have adequately addressed your comments raised in a previous round of review and you feel that this manuscript is now acceptable for publication, you may indicate that here to bypass the “Comments to the Author” section, enter your conflict of interest statement in the “Confidential to Editor” section, and submit your "Accept" recommendation.

Reviewer #1: All comments have been addressed

Reviewer #2: (No Response)

Reviewer #3: All comments have been addressed

2. Is the manuscript technically sound, and do the data support the conclusions?

Reviewer #1: Yes

Reviewer #2: Partly

Reviewer #3: Yes

3. Has the statistical analysis been performed appropriately and rigorously? 

Reviewer #1: N/A

Reviewer #2: Yes

Reviewer #3: Yes

4. Have the authors made all data underlying the findings in their manuscript fully available?

Reviewer #1: Yes

Reviewer #2: Yes

Reviewer #3: Yes

5. Is the manuscript presented in an intelligible fashion and written in standard English?

Reviewer #1: Yes

Reviewer #2: No

Reviewer #3: Yes

6. Review Comments to the Author

Reviewer #1: All the comments in the previous review have been sufficiently addressed, and the quality of the paper has been improved.

Reviewer #2: After carefully reading author responses and changes made in the revised manuscript. I have the following suggestions to improve the paper.

- Please add a concise explanation for Figure 2.

- Please improve the presentation and language throughout the paper.

Thank you

Reviewer #3: This study presents an improvement of the standard IEEE 802.15.4 CSMA/CA algorithm with consideration for data prioritization of heterogeneous data generated by WSN. The proposed CoSTP-MAC scheme implements data prioritization by classifying heterogeneous data into HPD and LPD with priority tags; CS 0 and 1 respectively. The traditional MAC frame format of the IEEE 802.15.4 was modified to accommodate data prioritization, with improved beacon called Beacon Plus Plus (B++). Subsequently, the traditional binary exponential backoff algorithm was also modified to obtain unique backoff algorithms peculiar to the data priority status. This implementation achieved fairness and improvement on how nodes gain access to the shared medium of transmission with respect to their priority status. Validating the proposed scheme, results show that HPD performed better than LPD by gaining quicker access to the shared transmission medium with improved performance compared to LPD. Subsequently, CoSTP-MAC outperformed other schemes with respect to PDR, throughput, latency, radio duty cycle and power consumption with improved operational longevity.

Overall, the paper presents a reasonable scientific contribution. The paper is now well-structured, and the ideas are properly presented. Moreover, the suggested revisions have been properly incorporated.

7. PLOS authors have the option to publish the peer review history of their article (what does this mean?). If published, this will include your full peer review and any attached files.

Reviewer #1: No

Reviewer #2: No

Reviewer #3: No

---

## [Editor Report · Acceptance letter]

27 Jul 2020

PONE-D-20-07077R1 

Optimized backoff scheme for prioritized data in wireless sensor networks: a class of service approach 

Dear Dr. Onwuegbuzie:

I'm pleased to inform you that your manuscript has been deemed suitable for publication in PLOS ONE. Congratulations! Your manuscript is now with our production department. 

Kind regards, 

on behalf of

Dr. Chi-Tsun Cheng 

Academic Editor

PLOS ONE